



# One-year characterization of organic aerosol composition and sources using an extractive electrospray ionization time-of-flight mass spectrometer (EESI-TOF)

Lu Qi[1,2], Alexander L. Vogel[1,a], Sepideh Esmaeilirad[1], Liming Cao[3], Jing Zheng[4], Jean-Luc Jaffrezo[5],
5   Paola Fermo[6], Anne KaspEr-Giebl[7], Kaspar R. Daellenbach[b], Mindong Chen[2], Xinlei Ge[2], Urs
Baltensperger[1], André S. H. Prévôt[1], Jay G. Slowik[1]

[1]Laboratory of Atmospheric Chemistry, Paul Scherrer Institute (PSI), 5232 Villigen, Switzerland

[2]Collavorative Innovation Center of Atmospheric Environment and Equipment Technology, Nanjing University of
Information Science & Technology, Nanjing, 210044, China

10   [3]Key Laboratory for Urban Habitat Environmental Science and Technology, School of Environment and Energy, Peking
University Shenzhen Graduate School, Shenzhen, 518055, China

[4]State Key Joint Laboratory of Environmental Simulation and Pollution Control, College of Environmental Sciences and
Engineering, Peking University, Beijing 100871, China

[5]University Grenoble Alpes, CNRS, IGE, 38000 Grenoble, France

15   [6]Department of Chemistry, University of Milan, 20133 Milan, Italy

[7]Institute of Chemical Technologies and Analytics, Vienna University of Technology, 1060 Vienna, Austria

[a]now at: Institute for Atmospheric and Environmental Sciences, Goethe University, Frankfurt am Main, Germany

[b]now at: Institute for Atmospheric and Earth System Research, University of Helsinki, Finland

*Correspondence to:* Jay G. Slowik (jay.slowik@psi.ch) and Andre Prévôt (andre.prevot@psi.ch)





**Abstract.**

The aerosol mass spectrometer (AMS), combined with statistical methods such as positive matrix factorization (PMF), has greatly advanced the quantification of primary organic aerosol (POA) sources and total secondary organic aerosol (SOA) mass. However, the use of thermal vaporization and electron ionization yields extensive thermal decomposition and ionization-induced fragmentation, which destroy chemical information needed for SOA source apportionment. The recently developed extractive electrospray ionization time-of-flight mass spectrometer (EESI-TOF) provides mass spectra of the organic aerosol fraction with a linear response to mass and no thermal decomposition or ionization-induced fragmentation. However, the costs and operational requirements of online instruments make their use impractical for long-term or spatially dense monitoring applications. This challenge was overcome for AMS measurements by measuring re-nebulized water extracts from ambient filter samples. Here, we apply the same strategy for EESI-TOF measurements of 1 year of 24-hour filter samples collected approximately every 4$^{th}$ day throughout 2013 at the NABEL monitoring station at Zurich-Kaserne, an urban site. The nebulized water extracts were measured simultaneously with an AMS. The application of positive matrix factorization (PMF) to EESI-TOF spectra resolved seven factors, which describe water-soluble OA: less and more aged biomass burning aerosol (LABB$_{EESI}$ and MABB$_{EESI}$, respectively), cigarette smoke-related organic aerosol (CS-OA$_{EESI}$), primary biological organic aerosol (PBOA$_{EESI}$), biogenic secondary organic aerosol (BSOA$_{EESI}$), and a summer mixed oxygenated organic aerosol (SMOA$_{EESI}$) factor. Seasonal trends and relative contributions of the EESI-TOF OA sources were compared with AMS source apportionment factors, measured water-soluble ions, cellulose, and meteorological data. Cluster analysis was utilized to identify key factor-specific ions based on PMF. Both LABB and MABB contribute strongly during winter. LABB is distinguished by very high signals from $C_6H_{10}O_5$ (levoglucosan and isomers) and $C_8H_{12}O_6$, whereas MABB is characterized by a large number of $C_xH_yO_z$ and $C_xH_yO_zN$ species two distinct populations: one with low H:C and high O:C, and the other with high H:C and low O:C. Two oxygenated summertime SOA sources were attributed to terpene-derived biogenic SOA, a major summertime aerosol source in Central Europe. Furthermore, a primary biological organic aerosol factor was identified, which was dominated by plant-derived fatty acids and correlated with free cellulose. The CS-OA factor contained a high contribution of nicotine and high abundance of organic nitrate ions with low *m/z*.

**1 Introduction**

Organic aerosol (OA) has significant but highly uncertain effects on climate and human health. OA is either directly emitted (primary organic aerosol, POA) or formed in the atmosphere by gas-phase oxidation of anthropogenic and natural volatile organic compounds, followed by condensation or nucleation of less volatile products (secondary organic aerosol, SOA). The Aerodyne aerosol mass spectrometer (AMS) provides online measurements of OA composition and in combination with statistical methods such as positive matrix factorization (PMF) has greatly advanced the quantification of primary organic aerosol (POA) sources and total secondary organic aerosol (SOA) mass (Jimenez et al., 2003; DeCarlo et al., 2008; Lanz et al., 2007; Ulbrich et al., 2009; Crippa et al., 2013a; Elser et al., 2016; Zhang et al., 2011). However, the AMS cost and





operational requirements make its use impractical for long-term or spatially dense monitoring applications. A few solutions were developed to overcome these shortcomings, e.g. the robust, less expensive, Aerosol Chemical Speciation Monitor (ACSM, Ng et al., 2011) and the Time of Flight (TOF)-ACSM were developed for long-term campaigns (Fröhlich et al., 2013, 2015), however, the low mass resolution of these instruments reduces their utility. Traditional offline techniques like gas
chromatography-mass spectrometry (GC-MS) or liquid chromatography-mass spectrometry (LC-MS) are chemically highly specific, but measure only a fraction the total organic aerosol. Another solution to this problem is the application of online instrumentation to extracted and re-aerosolized material from particle filter samples routinely collected at ambient monitoring stations (Daellenbach et al., 2016). Compared to online measurements, there are a few advantages: (1) practicality of long-term measurements; (2) practicality of making simultaneous measurements across multiple sites (possibly including sites
where access or infrastructure restrictions make the deployment of high-end instrumentation challenging); (3) such multi-site measurements can be performed with not only the same instrument type but actually a single instrument, improving comparability; (4) capability of particle composition measurement outside the size-dependent transmission range of the measuring instrument (e.g. coarse-mode particles in the AMS). On the other hand, drawbacks of the filter sampling and offline measurement strategy include possible positive / negative artifacts due to condensation/evaporation of semi-volatile organics
or aging during sampling, while compound dependent extraction efficiencies makes quantification more challenging.

The general analytical strategy outlined above, specifically the application of online instrumentation capable of highly time-resolved measurements to offline analysis of collected samples, has two key advantages relative to traditional offline techniques. First, the entire OA fraction can be analysed in comparison to the extracted one for the off-line analysis (64 %-76 % in the case of Switzerland (Daellenbach et al., 2016)). Second, sources that are tightly correlated on the 24-hr timescales
typical of filter measurement techniques may be more easily resolved at higher time-resolution; real-world source profiles from online measurements can therefore be used in advanced factor analysis of offline techniques to improve source separation (Daellenbach et al., 2016; Bozzetti et al., 2017).

While the offline-AMS technique has proven successful in characterizing POA sources and SOA mass, the AMS chemical resolution is limited by substantial thermal decomposition and ionization-induced fragmentation of the analyte molecules. This
problem is especially severe for the highly oxygenated, multifunctional molecules prevalent in SOA, and in most cases, prevents identification of source-specific SOA factors. In contrast, the recently developed extractive electrospray ionization time-of-flight mass spectrometer (EESI-TOF) is capable of online measurements at high time resolution without thermal decomposition or ionization-induced fragmentation (Lopez-Hilfiker et al., 2019). The EESI-TOF has been successfully deployed in several laboratory (Pospisilova et al., submitted) and field (Qi et al., 2019; Stefenelli et al., 2019) campaigns. It
yields signals that are linear with mass (albeit with ion-dependent response factors), making it suitable for source apportionment.

Here we present the first offline-EESI-TOF source apportionment analysis, conducted on one year of $PM_{10}$ filter samples collected in Zurich, Switzerland, and complemented by AMS measurements. This analysis describes the sources and processes governing the water-soluble fraction of OA. The chosen site is very well characterized, with multiple source apportionment



studies by online measurements with an AMS in different seasons (Lanz et al. 2010), an ACSM during a full year (Canonaco et al., 2013; Canonaco et al., 2015), and an EESI-TOF during summer and winter (Qi et al., 2019; Stefenelli et al., 2019), as well as offline measurements with an AMS and [14]C analyses (Daellenbach et al., 2016, 2017; Zotter et al., 2014).

**2 Methods**

**2.1 Site description and sample collection**

Sampling was conducted at the NABEL station in Zurich (47 º 22 'N, 8 º 33 'E), described in detail elsewhere (Canonaco et al., 2013; Lanz et al., 2007). Briefly, this station is an urban location, situated in the Kaserne Park in the city centre. In addition to sources characteristic of urban areas, local influences from nearby restaurants, traffic, and human activities (e.g. cigarette

smoking) are sometimes observed (Qi et al., 2019; Stefenelli et al., 2019). Meteorological parameters including temperature, relative humidity (RH), wind speed (WS), wind direction (WD), and global radiation, as well as concentrations of gas-phase species, including sulfur dioxide ($SO_2$), nitrogen dioxide ($NO_2$), and nitrogen oxide (NO) are recorded by the monitoring station.

$PM_{10}$ samples (91 filters) were collected every fourth day for 24 h throughout the year 2013 on quartz fiber filters (14.7 cm

diameter) using high-volume samplers (500 L min[-1]). Before sampling, the filters were pre-baked at 800 °C for 8 h. After collection, filters were wrapped in aluminum foil or lint-free paper, sealed in polyethylene bags and stored at -20 °C. Field blanks were collected and stored following the same procedure (Bozzetti et al., 2017; Daellenbach et al., 2017).

**2.2 Offline measurements**

The filters used for the present analysis were investigated by offline-AMS PMF in a previous study (Daellenbach et al., 2017).

Here, to optimize comparison between the offline-AMS and offline-EESI-TOF techniques, we produced a new aerosol extract, which was then nebulized for new simultaneous AMS and EESI-TOF measurements. In this way, we avoided differences due to extraction or nebulizer performance, filter aging during storage, system background / contamination, handling artifacts, etc., which might occur if the current EESI-TOF analyses were to be compared with the original offline-AMS study. As a consequence, the AMS dataset presented here is not completely identical to that of Daellenbach et al. (2017), although the

observed differences are small.

For each analyzed filter sample, one 16-mm diameter filter punch was subjected to ultrasonic extraction in 10 ml of ultrapure water (18.2 MΩ cm at 25 °C, total organic carbon (TOC) < 3 ppb) for 20 min at 30 °C. The extracted liquid was then filtered with 0.22 μm nylon membrane syringe filters and nebulized in synthetic air (80 % volume $N_2$, 20 % volume $O_2$; Carbagas, Gümligen, CH-3073 Switzerland) using a customized Apex Q nebulizer (Elemtental Scientific Inc., Omaha, USA) operating

at 60 °C. The resulting droplets were dried using a Nafion dryer and then analyzed by an extractive electrospray ionization





time-of-flight mass spectrometer (EESI-TOF) and a high-resolution time-of-flight AMS (HR-ToF-AMS). Total measurement time of each sample was 5 min. Before and after each sample, a measurement blank was generated by sampling nebulized ultrapure water for 10 min. Field blanks were measured following the same extraction procedure as the collected filter samples, yielding a signal not statistically different from that of nebulized ultrapure water. Each blank sample was recorded for 480 s.

**2.2.1 Offline AMS analysis**

The offline AMS analysis followed the methodology developed by Daellenbach et al. (2016). The offline-AMS operation was similar to other AMS measurements (Hu et al., 2013;DeCarlo et al., 2006). HR-TOF-AMS data was processed using the software SQUIRREL (SeQUential Igor data RetRiEvaL; D. Sueper, University of Colorado, Boulder, CO, USA) v.1.57 and PIKA (Peak Integration by Key Analysis) v.1.16 for the IGOR Pro software package (Wavemetrics, Inc., Portland, OR, USA). The high resolution mass spectral analysis was performed for each *m/z* (mass to charge) in the range of 12-120 at AMS V-mode and yielded a dataset consisting of 257 ions (excluding isotopes and $CO_2$-dependent ions). The interference of $NH_4NO_3$ on the $CO_2^+$ signal was corrected (Pieber et al., 2016) as follows:

$$CO_{2,real} = CO_{2,meas} - \left(\frac{CO_{2,meas}}{NO_{3,meas}}\right)_{NH4NO3,pure} \cdot NO_{3,real} \tag{1}$$

Where the $\left(\frac{CO_{2,meas}}{NO_{3,meas}}\right)_{NH4NO3,pure}$ correction factor was determined based on measurement period and varied between 1% and ~5%.

The AMS data were rescaled to the ambient concentration by normalizing the measured signal to the estimated water-soluble organic matter (WSOM) concentration, which was calculated as the product of the measured WSOC multiplied by the OM : OC ratios determined from the offline-AMS results. This method and the associated uncertainties are described in detail by Daellenbach et al. (2016; 2017). Note that because we do not attempt to correct for the water extraction efficiency of OM components, the analysis presented herein describes the source apportionment of AMS WSOM.

**2.2.2 Extractive Electrospray Ionization Time-of-flight Mass Spectrometer (EESI-TOF)**

The EESI-TOF is discussed in detail elsewhere (Lopez-Hilfiker et al., 2019;Qi et al., 2019), and a brief overview is presented here. Aerosol particles are continuously sampled through a 6 mm outer diameter (OD), 5 cm long multi-channel extruded carbon denuder housed in a stainless steel tube. The particle flow intersects a spray of charged droplets generated by a conventional electrospray capillary. Particles collide with the electrospray droplets and the soluble components are extracted. The droplets are evaporated gently, yielding ions that are detected by the TOF-MS. The electrospray working solution is a 50 / 50 water / methanol (MeOH, UHPLC-MS, LiChrosolv, mixture with 100 ppm NaI (99 %, Sigma-Aldrich) as a charge carrier. Organic components are detected as adducts with $Na^+$. Spectra are recorded in positive ion mode at 5 s time resolution. In normal operation, the EESI-TOF alternates between direct sampling of aerosol and sampling through a particle filter, to provide



a background measurement, however, the filter was not used in this study. Instead, the measurement blanks (nebulized ultrapure water) were used to determine the background. The EESI-TOF data was processed in Tofware 2.5.7 (Tofwerk AG, Thun, Switzerland).

We report the signal measured by the EESI-TOF in terms of mass flux of ions to the microchannel plate detector (attograms

$s^{-1}$ (ag $s^{-1}$), neglecting the mass of $Na^+$), calculated as shown in Eq. (2).

$$M_x = I_x \cdot (MW_x - MW_{cc}) \tag{2}$$

Here $M_x$ is the mass flux of ions united in ag $s^{-1}$, $x$ represents the measured molecular composition. $I_x$ is the recorded signal measured by EESI-TOF. $MW_x$ and $MW_{cc}$ represent the molecular weight of the ion and the charge carrier (e.g. $Na^+$, $H^+$), respectively. Note that this measured mass flux can in principle be related to the ambient OA concentration by the instrument

flow rate, EESI extraction / ionization efficiency, declustering probability, and ion transmission, where several of these parameters are ion-dependent and currently unknown (Lopez-Hilfiker et al., 2019). The EESI-TOF data was normalized to WSOC by using the AMS OM:OC ratios mentioned above. Similar to the AMS, no corrections for the efficiency of the water extraction from the filter samples is applied, and the offline EESI-TOF analysis therefore strictly relates only to the WSOM fraction. Note that because online EESI-TOF operation already requires extraction into the spray droplets (1:1

water:acetonitrile mixture, that major differences between the measured OA fraction between online and offline analyses are unlikely. A comparison of the EESI-TOF to the AMS signal in terms of total signal or mass, bulk properties, and source apportionment results is presented in Section 3.4.

### 2.2.3 Other offline measurements

Organic and elemental carbon (OC, EC) were determined using a thermo-optical transmission method with a Sunset OC-EC

analyzer, following the EUSAAR-2 thermal-optical transmission protocol (Cavalli et al., 2010). Water-soluble organic carbon was measured with water extraction followed by catalytic oxidation, and nondispersive infrared detection of $CO_2$ using a total organic carbon analyzer. Water-soluble major ions ($K^+$, $Na^+$, $Mg^{2+}$, $Ca^{2+}$, and $NH_4^+$ and $SO_4^{2-}$, $NO_3^-$, and $Cl^-$ and methane sulfonic acid) were determined using ion chromatography (Cuccia et al., 2013). Levoglucosan measurements (Piazzalunga et al., 2013) were performed with a high-performance anion exchange chromatographer (HPAEC) with pulsed amperometric

detection (PAD) using an ion chromatograph (Dionex ICS-1000). Free cellulose was determined using an enzymatic conversion to D-glucose and subsequent determination of glucose with an HPAEC.

### 2.3 Source apportionment techniques

The EESI-TOF PMF input data matrices included 91 filter samples. The input errors $\sigma_{ij}$ were calculated as the uncertainty related to ion counting statistics and ion-to-ion signal variability at the detector ($\delta_{i,j}$), added in quadrature to the blank variability

background ($\beta_{ij}$) (Qi et al., 2019).


$$\sigma_{ij} = \sqrt{{\delta_{ij}}^2 + {\beta_{ij}}^2} \qquad\qquad (3)$$

We applied a minimum error corresponding to the measurement of 1 ion during the 5 s averaging period. Variables with low signal-to-noise (SNR<0.2) were removed, whereas "weak" variables (0.2<SNR<2) were downweighted by a factor of 3 rather than 2 (following the recommendations of Paatero and Hopke (2009)). In total, 1070 fitted ions (1068 adducts with Na$^+$ and 2

with H$^+$) between $m/z$ 135 and 400 were identified.

The PMF source apportionment technique requires as input the time-series of ions from high-resolution mass spectral fitting along with their corresponding uncertainties. As for the EESI-TOF, the input AMS errors for PMF were calculated as the sum in quadrature of the AMS instrument uncertainties (including ion counting statistics and ion-to-ion signal variability at the detector ($\delta_{i,j}$)) and the blank variability ($\beta_{ij}$) (Ulbrich et al., 2009).

The offline-EESI-TOF and offline-AMS source apportionment was performed using positive matrix factorization (PMF) (Paatero and Tapper, 1994) as implemented by the Multilinear Engine (ME-2) and with model configuration and analysis executed via the SoFi (Source Finder, version 6.39) interface (Canonaco et al., 2013). PMF is a linear statistical model to describe the variability of a multivariate dataset. Specifically, an input data matrix (with elements $x_{i,j}$, where the $i$ and $j$ indices denote time and $m/z$, respectively) is described as the linear combination of a set of static factor profiles ($f_{i,k}$, where the $k$ index

denotes discrete factors) and temporal variation ($g_{k,j}$), as shown in Eq. (4):

$$x_{i,j} = \sum_{z=1}^{p}(f_{i,k} \cdot g_{k,j}) + e_{i,j} \qquad\qquad (4)$$

Here, $e_{i,j}$ represent elements of the residual matrix and $p$ is the total number of factors. The PMF algorithm iteratively solves Eq. (4) by minimizing the objective function Q, defined in Eq. (5):

$$Q = \sum_i \sum_j \left(\frac{e_{ij}}{\sigma_{ij}}\right)^2 \qquad\qquad (5)$$

$\sigma_{i,j}$ represents entries in the input uncertainty matrix.

The ME-2 implementation of the PMF algorithm offers an efficient exploration of the solution space by allowing the model to be directed towards environmentally meaningful rotations. Here this was done by constraining the factor profile elements $f_{i,k}$ for one or more factors (Canonaco et al., 2013), implemented using the $a$-value method, where the output $f_{i,k}$ for each constrained factor is required to satisfy Eq. (6):

$$f_{i,k} = f'_{i,k} \pm a \cdot f'_{i,k} \qquad\qquad (6)$$

Here $f'_{i,k}$ represents a predetermined anchor profile, and the scalar $a$ ($0 \le a \le 1$) determines the tightness of constraint. Anchor profiles may be obtained by several methods, including prior studies, laboratory measurements of known sources, or analysis of a subset of the current dataset, and are discussed on a case-by-case basis in Sections 3.1 (AMS) and 3.2 (EESI-TOF) (Canonaco et al., 2015).



### 2.4 Identification of source-specific ions

To determine ions characteristic of individual factors (or groups of related factors), agglomerative hierarchical clustering was conducted on the EESI-TOF matrix of PMF profiles. A dendrogram is used to show relationships between members of a group. A more detailed description is found in Qi et al. (2019).

Here, we summarize the steps: (1) Calculation of the standardized value (z-score) along the ions by using Eq. (7):

$$z = \frac{x - \mu}{\sigma} \tag{7}$$

The $\mu$ is the mean value, $\sigma$ is the standard deviation, $Z$ represents the distance between the raw score and the mean value in units of the standard deviation. (2) Formation of groups of the new calculated data by using the Euclidean distance (Eq. 8):

$$dist\left(x_i, x_j\right) = \sqrt{\sum_{m=1}^{n}\left(\frac{x_{im} - x_{jm}}{\sigma_m}\right)^2} \tag{8}$$

Here, $i = (1,\ldots,m)$, $j = (1,\ldots,m)$. (3) clustering along the columns (producing row-clustered groups of factor), and along the rows (producing the clustered ions to each group). The calculation and the generation of the dendrogram were performed with Matlab R2017b.

## 3 Results and discussions

### 3.1 Interpretation of AMS-PMF factors

Here we summarize the results of the AMS-PMF analysis on the WSOM fraction, which as noted in Section 2.2 are very similar to those of Daellenbach et al. (2017), conducted on different extracts from the same ambient filter samples. HOA$_{AMS}$ and COA$_{AMS}$ mass profiles were constrained using anchor profiles obtained from winter in Paris (Crippa et al., 2013b) with $a$-values of 0.1 and 0.2, respectively. A six-factor solution was selected as the best representation for the AMS PMF analysis,
yielding factors identified as hydrocarbon-like OA (HOA$_{AMS}$), cooking OA (COA$_{AMS}$), biomass burning OA (BBOA$_{AMS}$), winter oxygenated OA (WOOA$_{AMS}$), summer oxygenated OA (($_{AMS}$), and sulfur-containing OA (SCOA$_{AMS}$). The methods of factor classification and factor selection for the AMS PMF results are similar to Daellenbach et al. (2017), although a detailed sensitivity analysis was not repeated. Figures 1 and S1 show the mass spectra and the time series of the AMS factors, respectively. The main characteristics of the AMS PMF factors are summarized below. BBOA$_{AMS}$ exhibits high contributions
from C$_2$H$_4$O$_2^+$ ($m/z$ 60), a characteristic ion from the fragmentation of anhydrosugars such as levoglucosan (Sun et al., 2013; Takahama et al., 2013; Lin et al., 2016). The BBOA$_{AMS}$ time series shows the expected seasonal variation with elevated concentrations in winter, supporting the identification of this factor. The oxygenated OA factors are resolved based on the differences in their seasonal behavior: $_{AMS}$ (elevated in summer) and WOOA$_{AMS}$ (elevated in winter). This season-based separation of OOA factors is typical of offline AMS analysis (Bozzetti et al., 2016; Daellenbach et al., 2017), but contrasts
with typical results from PMF analysis of highly time-resolved data from short-term measurements, where OOA factors are



more likely to be separated by volatility and / or photochemical age (Zhang et al., 2011; Jimenez et al., 2009). Even though $_{AMS}$ has a high contribution in summer and shows an increase with rising temperature, it also contributes, to a lesser degree, during the other seasons (Fig. S1, (Daellenbach et al., 2017)). The mass spectrum of SCOA$_{AMS}$ is dominated by the fragment $CH_3SO_2^+$, which was found to derive from a sulfur-containing compound other than methanesulfonic acid (MSA) (Daellenbach

et al., 2017). This factor is believed to derive from primary traffic related sources, and in size-resolved analyses at other sites it has been found mainly in the coarse mode (Vlachou et al., 2018). The meteorological data, ions data, and the factor comparison between EESI-TOF and AMS are presented in Section 3.2.2.

**3.2 EESI-TOF source apportionment**

**3.2.1 EESI-TOF solution selection**

Selection of an appropriate number of factors is a critical component of any PMF analysis. Increasing the number of factors gives the model more freedom to explain subtle variations of the data, but too many factors may force the model to split a physically meaningful factor into non-meaningful ones. In this section, we present how we selected the number of PMF factors based on the residual analysis and the solution interpretability. The offline EESI-TOF PMF analysis was performed for solutions with 1 to 10 factors. Solutions were assessed based on the internal consistency of the factor mass spectra, and

comparison of factor time series with offline-AMS PMF solutions, external tracers and auxiliary data. The $Q$ normalized by its expected value ($Q / Q_{exp}$) between the various runs was around 2.4 for the six-factor solution and higher (Fig. S2), Here we present a brief overview of the retrieved solutions as a function of the number of factors. Characteristics of the factors, including justifications for their assigned labels, are presented in Section 3.2.2.

The 5-factor solution is largely driven by differences between the winter and summer seasons (Fig. S3). The solution includes

two factors related to biomass combustion. A less aged biomass burning (LABB$_{EESI}$) factor, dominated by the ion $m/z$ 185.04 ($C_6H_{10}O_5Na^+$, ($C_6H_{10}O_5$, levoglucosan and its isomers). In the following text, the neutral formula is used to represent ions), exhibits high contributions over the last few months of the year, while a more aged biomass burning (MABB$_{EESI}$) is elevated during both winters. These two factors are distinguished by their mass spectra, as discussed further in Section 3.2.2. A biogenic secondary organic aerosol (BSOA$_{EESI}$) factor contributes during the warm season and has a negligible contribution from

$C_6H_{10}O_5$. The primary biological organic aerosol (PBOA$_{EESI}$) factor has a different time series that has no correlation with other external tracers. The last factor seems to be mixed due to the two major peaks at $m/z$ 163.12 and $m/z$ 185.04. Based on the unique ion of $m/z$ 163.12 in the factor mass spectrum, which is tentatively explained by nicotine ($C_{10}H_{15}N_2^+$), we denote it here as the "163.12" factor.

In the 6-factor solution (Fig. S4), the LABB$_{EESI}$, MABB$_{EESI}$, BSOA$_{EESI}$, and PBOA$_{EESI}$ factors are qualitatively similar to their

counterparts in the 5-factor solution. However, the "163.12" factor is separated into a cigarette smoke-related OA (CS-OA$_{EESI}$) factor retaining the prominent peak at 163.12 and a less aged biomass burning factor (LABB2$_{EESI}$) with a strong contribution from $C_6H_{10}O_5$ and a high correlation with BBOA$_{AMS}$.



Increasing the number of factors to 7 yields an additional factor, described as summer oxygenated organic aerosol (SMOA$_{EESI}$), which exhibits a high peak in summer but also has a significant contribution throughout the year (Fig. 2). The time series correlates with $_{AMS}$, and the profile is similar to that of photochemically generated, biogenic-dominated SOA identified from online measurements at the same site (Stefenelli et al., 2019), as discussed in Section 3.2.2. As discussed in section 3.3,

SOAA$_{EESI}$ evidences a more regional/mixed composition than BSOA$_{EESI}$.

When eight factors are assumed, the profile of the new factor points to an additional more aged biomass burning factor (MABB2$_{EESI}$) (Fig. S5) with two high peaks at $m/z$ 165.09 (C$_7$H$_{13}$NO$_2$) and $m/z$ 185.04 (C$_6$H$_{10}$O$_5$). Adding this factor alters the time series of other factors, decreasing their correlation with relevant tracer. Further, its time series has no clear seasonal trend or correlation with other reference, so it does not provide further source information and is therefore disregarded. Increasing

the number of factors beyond 8 yielded additional split or mixed factors without adding any new interpretable factors. We therefore selected the 7-factor solution for the analysis below.

### 3.2.2 Overview of EESI-TOF source apportionment

An overview of the EESI-TOF source apportionment analysis is presented in this section, with the factors discussed in detail in Section 3.3. Figure 2a shows the time series of the seven EESI-TOF PMF factors (LABB1$_{EESI}$, LABB2$_{EESI}$, CS-OA$_{EESI}$,

PBOA$_{EESI}$, MABB$_{EESI}$, BSOA$_{EESI}$, SMOA$_{EESI}$) over the entire year, together with relevant AMS PMF factors, meteorological conditions, and other ancillary measurements. The retrieved factors are analyzed in terms of their composition, correlation with markers and relationship to offline-AMS factors retrieved over the same period. Figure 2b presents the factor mass spectra, with ions colored by number of nitrogen.

Figures 3 to 5 show alternative representations of the factor mass spectra. Figure 3 presents the spectra as stacked histograms

according to the chemical family, binned by the number of carbon atoms, with the vertical axis representing the relative signal intensity. We define 3 chemical families: C$_x$H$_y$N$_z$O$_k$, C$_x$H$_y$N$_z$, and C$_x$H$_y$O$_z$, with the latter further separated into five groups by atomic H:C ratios: H:C <1.1, 1.1-1.3, 1.3-1.5, 1.5-1.7 and >1.7. The factor mass spectra are also presented as Van Krevelen plots (atomic H:C vs. O:C ratio) in Figs. S6 and S7 for the POA and SOA factors, respectively. Points are sized by the fraction of each ion apportioned to the respective factor and colored by the number of carbon atoms, except in the case of CS-OA$_{EESI}$,

where the color scale denotes the number of nitrogen atoms.

As evidenced from the previous section and Figs. 2 and 3, many of the dominant ions in the EESI-TOF PMF analysis are shared by multiple factors. Here, we utilize a cluster analysis to identify ions unique or nearly unique to a single factor or group of factors, as described in detail in Section 2.4. Figure 4 shows the results of this analysis as a clustergram. Colors denote the z-score of each factor / ion combination. Hierarchical agglomerative clustering was performed independently on (1) the z-

score profile of each ion across all factors (vertical axis) and (2) the factor profile across all ion z-scores (horizontal axis). The outcomes of these cluster analyses are represented as dendrograms on the vertical and horizontal axes, respectively. The ions are clustered based on having a similar z-score pattern across the factors and the resulting tree is shown on the left, colored subjectively to guide the eye. Clearly, the dendrogram divides the factors into three main groups: one group including CS-





$OA_{EESI}$, $PBOA_{EESI}$, and $LABB2_{EESI}$, a biomass burning group ($LABB1_{EESI}$ and $MABB_{EESI}$), and a biogenic OA group ($BSOA_{EESI}$ and $SMOA_{EESI}$). Key ions are defined as those having a z-score > 1.5 for a given factor. These ions are shown in Fig. 5 as stacked histograms binned by the number of carbon atoms, with colors denoting chemical family ($C_xH_yN$, $C_xH_yNO_z$, and $C_xH_yO_z$, with the latter further separated by the H:C ratio). The left column displays these ions in terms of their relative

intensity within each factor profile, while the right column shows the number of identified ions. A full list of the identified key ions is given in Table S1.

Overall, for all the EESI-TOF factors, the assigned ions exhibit systematic patterns supporting the above identification. Fig. 6a and 6b show the mass defect, defined as the exact $m/z$ minus the nearest integer $m/z$, as a function of $m/z$ for the uniquely assigned ions for the seven factors. For several displayed factors, linear correlations or clusters of points are observed. Figure

6a shows the majority of the distinguished molecules (defined as factor-specific ions) of $LABB1_{EESI}$ and $CS-OA_{EESI}$ factors spread tightly from $m/z$ 100 to 400, while the factor of $LABB2_{EESI}$ clusters from $m/z$ 300 to 400 with a few additional points from $m/z$ 150 to 200. The mass defects of the $LABB1_{EESI}$-factor-specific ions are lower than the $CS-OA_{EESI}$ - and $LABB2_{EESI}$-factor-specific ions, which indicates that there are more aromatic ions (with a lower H:C ratio) in the LABB1 factor. The slope for the $LABB1_{EESI}$ factor of $4.6*10^{-4}$ is consistent with addition of CH groups (theoretical slope $6*10^{-4}$). It is also consistent

with the slope of the primary biomass burning source from a Zurich field campaign (with a slope of $4.9*10^{-4}$, Qi et al., 2019). Here, the slope of the $CS-OA_{EESI}$ factor is $6.4*10^{-4}$, while the slope of the CS-OA factor from the Zurich field campaign is $8*10^{-4}$. As shown in Fig. 6b, the mass of the markers of the $PBOA_{EESI}$ factor spread from $m/z$ 250 to 400 with a high mass defect. A general trend is that the mass defect value of the $BSOA_{EESI}$ factor is a slightly higher than of the $SMOA_{EESI}$ factor. Both the slopes of $BSOA_{EESI}$ ($8.7*10^{-4}$) and $SMOA_{EESI}$ ($7.0*10^{-4}$) are consistent with the addition of CHO functional groups

(theoretical slope $= 1*10^{-3}$).

### 3.3 EESI-TOF source apportionment factors

*Cigarette smoke-related OA (CS-OA$_{EESI}$)*

The $CS-OA_{EESI}$ time series lacks a clear seasonal trend. However, as shown in Fig. 2a, it correlates strongly with the EESI-

TOF nicotine ion ($R=0.89$). As a reduced nitrogen compound, nicotine likely forms a stable ion by abstracting a hydrogen from water, leading to the observed cation. The stacked histogram of the CS-OA factor (Fig. 3) is unique among the resolved factors in having strong contributions from the CHN family. Other significant contributions come from $C_6H_{10}O_5$ and $C_8H_{12}O_6$ (Fig. 2b). As discussed above, these species are prevalent also in biomass combustion, and may occur in this factor due to combustion of biomass in the cigarette.

Oxidized nitrogen (ON) species ($C_xH_yO_zN_1$ and $C_xH_yO_zN_2$) are significant in the $CS-OA_{EESI}$ factor, as shown in Fig. S6c. It is only slightly oxygenated, with an O:C ratio below 0.2, and has a high H:C ratio of approximately 1.9. The field measurements at the same site had identified a cigarette smoke factor with a spectral fingerprint similar to $C_{10}H_{14}N_2$ (Qi et al., 2019; Stefenelli et al., 2019). As shown in Fig. 5, the factor-specific ions of the CS-OA factor from cluster analysis have high abundance of





CHNO, CHN and a high H:C ratio (> 1.5), which is consistent with our discussion that the factor is primary and dominated by the nitrogen-containing species.

### *Primary biological organic aerosol (PBOA$_{EESI}$)*

The mass spectrum of the PBOA$_{EESI}$ factor is shown in Fig. 2b. Strong contributions from slightly oxygenated ions with high carbon number and high H:C ratios, such as $C_{19}H_{32}O_3$, $C_{18}H_{30}O_3$, $C_{14}H_{22}O_3$, $C_{16}H_{26}O_2$, $C_9H_{16}O_2$, are consistent with fatty acids identified from plants (http://plantfadb.org) (Tervahattu et al., 2005; Schilling et al., 2016). As shown in Fig. 3, the overall mass spectrum of the PBOA$_{EESI}$ factor is shifted towards ions with higher carbon number (i.e., $C_{12}$ to $C_{20}$) relative to the other factors. Figure 4 shows that the ions with high *z*-score in PBOA$_{EESI}$ are mostly unique to this factor. These ions are

characterized by high carbon number and high H:C ratio, as shown in Fig. 5. Of all the factors, only LABB2$_{EESI}$ (discussed below) has unique ions with a comparably high carbon number distribution, however, the factor-specific ions of these two factors are not overlapping Fig. 5).

The PBOA$_{EESI}$ factor is observed throughout the year, with slightly higher contributions during summer (Fig. 2a, S9). PBOA typically consists of solid airborne particles derived from biological organisms, including microorganisms and fragments of

biological materials such as plant debris and animal dander (Després et al., 2012). PBOA has been observed previously as a significant source of coarse aerosol organic matter (OM, aerodynamic diameter > 2.5 μm) in Switzerland (Després et al., 2012; Bozzetti et al., 2016; Vlachou et al., 2018). The most frequently occurring biopolymer in terrestrial environments is cellulose, as around 50 % of dry weight cellulose is from leaves (Sánchez-Ochoa et al., 2007; Jaenicke, 2005). Atmospheric "free cellulose" has been determined as a proxy for plant debris. As shown in Fig. 2a, the time series of PBOA$_{EESI}$ is similar to the

one of cellulose (*R*=0.83), although the number of cellulose measurements is limited to only 12 filters. Nevertheless, like the factor mass spectrum this correlation is consistent with the assignment of this factor to PBOA.

We also considered cooking-related emissions as an alternative assignment for the PBOA$_{EESI}$ factor. Viewed broadly, these two emissions sources are somewhat similar in that they both have strong contributions from fatty acids, which are the salient features of the PBOA$_{EESI}$ mass spectrum. Indeed, a COA$_{AMS}$ factor is resolved, but no COA is retrieved from the EESI-TOF

dataset despite previous studies identifying online cooking emissions in EESI-TOF data during summer and winter at the same site. However, offline-AMS analyses have previously shown COA to have a low extraction efficiency (Daellenbach et al., 2016), resulting in low contribution in the EESI-TOF dataset. Further, the highest COA$_{AMS}$ concentrations occur during the period of 14 April – 08 May, during which no EESI-TOF data is available. Without this strong temporal feature, COA may contribute too little to the variability of the EESI-TOF dataset to be resolved. A detailed comparison of the retrieved PBOA$_{EESI}$

profile with previously obtained EESI-TOF COA factors shows that dominant PBOA$_{EESI}$ ions are different from the major components of cooking-related EESI-TOF factors obtained from source apportionment of online summer and winter mass data, e.g. $C_{16}H_{30}O_3$, $C_{18}H_{34}O_2$ (Stefenelli et al., 2019; Qi et al., 2019). Further, we note that the time series of the PBOA$_{EESI}$



and COA$_{AMS}$ factors are not well correlated, suggesting different sources and thus a unique source for PBOA$_{EESI}$ unrelated to cooking emissions.

### *Less aged biomass burning factors (LABB1$_{EESI}$ and LABB2$_{EESI}$)*

Two factors were attributed to relatively fresh biomass burning emissions, denoted here as less aged biomass burning type 1 and 2 (LABB1$_{EESI}$ and LABB2$_{EESI}$, respectively). LABB1$_{EESI}$ displays many characteristics that are similar to primary or slightly aged wood burning emissions from previous EESI-TOF and AMS source apportionment studies. The time series of the LABB1$_{EESI}$ factor is correlated with the BBOA$_{AMS}$ factor ($R$=0.6, Fig. 2a). LABB1$_{EESI}$ also correlates with the $C_6H_{10}O_5$ ion measured by the EESI-TOF ($R$=0.43), corresponding to levoglucosan and its isomers, which are well-known tracers of biomass

combustion. LABB1$_{EESI}$ shows a pronounced yearly cycle with high concentration during both winters, consistent with previous studies identifying biomass burning as a major source of wintertime OA in Zurich and central Europe (Crippa et al., 2013a; Crippa et al., 2014; Bozzetti et al., 2016). The factor profile is dominated by the ions of $C_6H_{10}O_5$ and $C_8H_{12}O_6$ as shown in Fig. 2b, similar to fresh wood burning emissions resolved by source apportionment of online EESI-TOF data (Qi et al., 2019). Although the EESI-TOF provides only a molecular formula and not structural information, we note that the dominant

contribution of a very small number of ions (i.e., $C_6H_{10}O_5$ and $C_8H_{12}O_6$) to the factor profile suggests that these ions result from a process such as cellulose pyrolysis, which gives rise to a relatively small number of discrete major products (including levoglucosan) as opposed to oxidative processing, which is characterized by more complex branching pathways and thus a broader distribution of chemically related compounds (e.g. homologous ion series). As a result, the $C_8H_{12}O_6$ ion is likely also a pyrolysis product or other primary emission and not, for example, MBTCA (3-methyl-1,2,3, -butanetricarboxylic acid) which

is an important product of α-pinene oxidation. Figure 3 and S6 show that, in addition to the strong contributions from $C_6H_{10}O_5$ and $C_8H_{12}O_6$, LABB1$_{EESI}$ is unique among the retrieved factors in having a higher fractional signal from ions with low H:C value. This trend is amplified in the key ions identified from the clustergram analysis (Fig. 4), as shown in Fig. 5, where (aside from $C_6H_{10}O_5$ and $C_8H_{12}O_6$), the ions unique to LABB1$_{EESI}$ consist almost entirely of ions with H:C < 1.5. This contrasts sharply with the other factors, where ions with low H:C are rare. The low H:C may indicate more aromatic character because

combustion origins of primary OA and / or oxidation products of aromatics, which may include significant contribution from ring-opening reactions.

The LABB2$_{EESI}$ factor is enhanced during the second winter only, while concentrations during the first winter are indistinguishable from those during summer. Concentrations begin to increase in September, and continue increasing throughout the rest of the year. Because the filter samples were measured in random sequence, this does not reflect an artifact

of the offline-EESI measurement system, such as a drift in EESI-TOF performance or gradual contamination of the aerosol generation system. It is therefore likely that this time series represents a real feature of the aerosol composition. The scatter plots (Fig. S8) show that the sum of the LABB1$_{EESI}$ + LABB2$_{EESI}$ factors has a higher correlation with BBOA$_{AMS}$ than LABB1$_{EESI}$ alone. The mass spectrum of LABB2$_{EESI}$ is dominated by $C_6H_{10}O_5$ and $C_8H_{12}O_6$, with high contributions of C19-





C21 ions (14 % to the factor). This is demonstrated by the cluster analysis (Fig. 5, Table S1), which indicates that the factor-specific ions with C19 and C21 are predominantly in this factor. The Van Krevelen plot (Fig. S6b) shows that this factor is dominated by ions with a high H:C ratio (above 1.7, Fig. 5), low O:C ratio (below 0.25) and high carbon number, which is consistent with our identification that the factor is likely from primary emissions (Bertrand et al., 2018; Elser et al., 2016).

### More aged biomass burning OA (MABB$_{EESI}$)

The time series of MABB$_{EESI}$ and WOOA$_{AMS}$ are strongly correlated (Fig. 2a, $R$=0.85). Both also have a strong correlation with the secondary aerosol component NH$_4^+$ ($R = 0.7$). These correlations suggest that MABB$_{EESI}$ is significantly influenced by SOA. However, a strong contribution from C$_6$H$_{10}$O$_5$ is also observed, suggesting that the factor also contains primary

emissions, although this POA species comprises a substantially lower fraction of the total factor than in LABB1$_{EESI}$ and LABB2$_{EESI}$. As also shown in Fig. 2b, major components of the MABB$_{EESI}$ mass spectrum (include.g., C$_6$H$_{10}$O$_5$, C$_7$H$_8$O$_5$, C$_9$H$_{16}$O$_4$, and C$_8$H$_{12}$O$_6$) are similar to those in mass spectra of aged biomass burning emissions retrieved from smog chamber experiments (Bertrand et al., submitted) and MABB factors from source apportionment of online EESI-TOF data from a winter study in Zurich (Qi et al., 2019). Figure S11 shows the distinct low relative contribution of C$_6$H$_{10}$O$_5$ and C$_8$H$_{12}$O$_6$ ions to the

MABB$_{EESI}$ factor.

Figure 7 shows that the factor-specific ions of the MABB$_{EESI}$ factor are classified into two distinct populations (Fig. 6b), with lower H:C and higher O:C ratio on the one hand and higher H:C and lower O:C ratio on the other hand. These two populations are consistent with the stacked histogram of MABB shown in Fig. 5. The lower mass defect population with lower H:C ratio is consistent with phenol / cresol oxidation enriched with C6 / C7 ions which are known as important biomass burning SOA

precursors (Burns et al., 2016). The higher mass defect population with higher H:C ratios was composed of CHON and CHO group ions. Furthermore, phenol SOA has been shown to have a low relative response factor (RRF) in EESI-TOF, indicating an underestimation of these ions (Lopez-Hilfiker et al., 2019).

### Biogenic secondary organic aerosol (BSOA$_{EESI}$) and summer mixed oxygenated organic aerosol (SMOA$_{EESI}$)

Both BSOA$_{EESI}$ and SMOA$_{EESI}$ factors show elevated concentrations in summer and have negligible contributions from levoglucosan (C$_6$H$_{10}$O$_5$). The BSOA$_{EESI}$ factor exhibits a high contribution during warm seasons, spring and summer, but is near zero during winter (Fig. 2a). The time series of the SMOA$_{EESI}$ factor also shows an elevated contribution in summer, but differs from BSOA$_{EESI}$ in that it also has non-zero contribution in winter. Figure S10 (Fig. S12) shows the correlation of the

three factors with the ambient temperature. While the SMOA$_{EESI}$ factor does not show a clear dependency on temperature, BSOA$_{EESI}$ increases exponentially with temperature ($y = 1.2478e^{0.1581x}$), consistent with the known relationship for terpene emissions and biogenic aerosol in terpene-dominated regions. While SOOA$_{AMS}$ stems largely from biogenic precursors, this factor likely includes also a smaller proportion of compounds from other sources, whereas BSOA$_{EESI}$ represents rather pure biogenic SOA.





Figure 2b shows that the ions with the highest signal in BSOA$_{EESI}$ are C$_7$H$_{10}$O$_5$, C$_8$H$_{12}$O$_5$, C$_9$H$_{14}$O$_5$, C$_{10}$H$_{16}$O$_5$, and C$_{10}$H$_{16}$O$_6$, while other ions, i.e. C$_{14}$H$_{22}$O$_5$, C$_{15}$H$_{22}$O$_5$, C$_{15}$H$_{24}$O$_5$, C$_{15}$H$_{22}$O$_6$, are tentatively identified as sesquiterpene oxidation products. This differs slightly from SMOA$_{EESI}$, where C$_8$H$_{12}$O$_4$, C$_9$H$_{14}$O$_4$, C$_9$H$_{16}$O$_5$, C$_{10}$H$_{16}$O$_4$, and C$_{10}$H$_{16}$O$_5$ show the strongest signals. Figure 3 shows that the BSOA$_{EESI}$ factor contains more organic nitrogen species than the SMOA$_{EESI}$ factor. Figure S7b and

S7c show Van Krevelen plots for these two factors. BSOA$_{EESI}$ has a higher O:C ratio than SMOA$_{EESI}$ (1-1.2 vs. 0.4-0.6). The two factors are compared in more detail in Fig. 5, with the factor-specific ions. The range of H:C ratios is between 1.1 and 1.5 for the marker ions in both factors (except for the CHNO family). The carbon numbers of factor-specific ions in BSOA$_{EESI}$ factor are spread from C8 to C21. The high C numbers are consistent with the presence of sesquiterpene oxidation products and terpene dimers. The SMOA$_{EESI}$ factor mostly with less than 10 carbon atoms (C7, C8, C9 and C10) likely indicates

fragmentation products from terpene oxidation in the gas phase followed by condensation after oxidation of light aromatics via ring opening. This is consistent with our temperature comparison above that BSOA$_{EESI}$ factor is likely SOA from pure biogenic emissions, SMOA$_{EESI}$ factor is likely mixed and regional.

Figure 8 compares these factor mass spectra with a factor dominated by terpene SOA ("Daytime SOA2, Daytime SOA1") derived from PMF analysis of a summer field campaign at the same site in Zurich, as well as a mass spectrum from field

measurements during spring in Hyyttiala, Finland, located in the remote boreal forest (Stefenelli et al., 2019; Qi et al., 2019; Pospisilova et al., submitted). This comparison shows the BSOA$_{EESI}$ factor and the SMOA$_{EESI}$ factors to be qualitatively similar to terpene-derived SOA. Additionally, the terpene-derived biogenic SOA has already been identified as a major summertime aerosol source in Central Europe (Zhang et al., 2018; Claeys et al., 2007; Ng et al., 2007; Canonaco et al., 2015; Daellenbach et al., 2017).

### 3.4 EESI-TOF and AMS comparison

For the comparison of EESI-TOF and AMS results, no relative sensitivity corrections were applied to the EESI-TOF data, although it is known that compound-dependent differences exist (Lopez-Hilfiker et al., 2019). Figure 9a shows the total ion signal (ag s$^{-1}$) measured by the EESI-TOF as a function of the OA concentration measured by the AMS, with the points colored

by date. Agreement is generally good except during winter, where the ratio of EESI-TOF to AMS is lower. This corresponds to high fractional contributions from the EESI biomass burning factors, especially the SOA-dominated MABB$_{EESI}$. The apparently reduced EESI-TOF response is thus likely driven by the lower EESI-TOF sensitivity to SOA from light aromatics compared to terpenes (Lopez-Hilfiker et al., 2019). Figure 9b shows the mass flux of EESI-TOF SOA signal (MABB + BSOA + SMOA) as a function of AMS SOA mass, colored by date, while Fig. 9c shows the comparison between EESI-TOF POA

and AMS POA, correlating well with each other. Similar to Fig. 9a, the SOA-dominated period toward the winter exhibits a lower relative sensitivity for the EESI-TOF than the terpene-dominated summer season.

Figures 9d and 9e show the O:C and H:C atomic ratios, respectively, for the EESI-TOF versus those of the AMS. The estimated O:C ratios by the EESI-TOF (around 0.3-0.45, again with no ion-dependent response factors applied) are systematically higher



than those measured by the AMS (around 0.2-0.3). This bias is similar to that observed in online data for winter and summer aerosol in Zurich (Stefenelli et al., 2019; Qi et al., 2019). On the other hand, the H:C ratios of these two instruments show fair consistency with values around 1.5 for the EESI-TOF and AMS analyses. This is not consistent with our observation for summer and winter aerosol in Zurich and for aging experiments of wood burning emissions in an environmental chamber,

where all measured H:C ratios were higher for the EESI-TOF than for the AMS. Note also that, unlike the comparison of total EESI-TOF and AMS signals, there are no seasonally-dependent differences in the measured H:C or O:C ratio.

Figure 10 shows the stacked time series of the AMS PMF factors and EESI-TOF PMF factors. Also shown are pie charts of the mean EESI-TOF factor contributions (Fig. 10c) and the mean AMS factor contributions (Fig. 10d) over the entire measurement period. As discussed earlier, this apportionment specifically describes the WSOM fraction, as no WSOM-to-OM

correction factors are applied.

Overall, the sum of the primary factors of $LABB1_{EESI}$ (12 %) and $LABB2_{EESI}$ (6.5 %) contributes 18.5 % of the EESI-TOF signal and compares with the $BBOA_{AMS}$ factor (12 %). The fraction of secondary $MABB_{EESI}$ (20.3 %) factor is a bit lower than the $WOOA_{AMS}$ factor (22 %). The source of $CS-OA_{EESI}$ contributes to 9.3 %, which must have contributions from some AMS factors, e.g. from wood burning-related and cooking. The secondary factors $BSOA_{EESI}$ (19.7 %) and $SMOA_{EESI}$ (9.9 %)

contribute 29.6 % of the EESI-TOF signal compared to 37.6 % of the total apportioned mass for the AMS summer factor SOOA. The $PBOA_{EESI}$ factor exhibits the strongest difference, with 22.3 % in the EESI-TOF, while PBOA is not resolved at all in the AMS. Daellenbach et al. (2017) did not separate a PBOA factor in their AMS PMF analysis, neither unconstrained nor using the mass spectral signature from Bozzetti et al. (2016). Three methods (based on factor profiles, coarse OC and cellulose) were used to estimate the influence of PBOA in Bozzetti et al. (2016), reporting that offline measurement is with a

factor of 3 to 10 times lower PBOA in the warm season. Here, the EESI-TOF measurement shows the advantage for measuring the samples at molecular level, enabling the separation of $PBOA_{EESI}$ and $CS-OA_{EESI}$ factors from PMF analysis.

## 4 Conclusions

In this study, we analyzed 86 filters collected at the NABEL monitoring station at Zurich Kaserne, an urban background site. These filters were collected for 24 hours each, approximately every 4th day throughout 2013, then measured by utilizing the

offline-AMS method (water extraction followed by re-nebulization and measurement) to the EESI-TOF. It is the first offline work to characterize the secondary organic aerosol sources and composition using a new developed instrument extractive electrospray ionization time-of-flight mass spectrometer (EESI-TOF). The increased chemical specificity of the EESI-TOF allows for additional, meaningful factors to be resolved relative to the AMS.

Positive matrix factorization (PMF) analysis was conducted on the offline-EESI-TOF data, yielding seven factors describing

water soluble organic material (WSOM): two less aged biomass burning factors ($LABB1_{EESI}$ and $LABB2_{EESI}$) indicating a strong aromatic influence; cigarette smoke organic aerosol ($CS-OA_{EESI}$, characterized by the contribution from nicotine); primary biological organic aerosol ($PBOA_{EESI}$) identified by fatty acids from plants; more aged biomass burning ($MABB_{EESI}$)



characterized by the key feature from wood burning chamber measurement, biogenic secondary organic aerosol ($BSOA_{EESI}$) and summer mixed oxygenated organic aerosol ($SMOA_{EESI}$) showing enhanced contribution from ions characteristic of monoterpene oxidation. The offline EESI-TOF PMF retrieved a $PBOA_{EESI}$ factor, separated less aged and more aged factors from biomass burning, and presented winter and summer dominated emissions respectively, features that are not possible for

AMS PMF analysis. We performed cluster analysis of the EESI-TOF ions followed by correlation with the resolved factors, which identifies factor-specific ions of each factor. These characteristic ions represent potential markers for future studies.

Overall, the EESI-TOF analysis was supported and corroborated by the AMS PMF analysis. This work highlights the potential of offline, highly chemically-resolved data provided by an EESI-TOF, for identification of the key sources over a long time

period.

**Data availability.**

The data are available under ACP version.

**Author Contribution.**

LQ was the main author. LQ, AV, SE, LC, JZ, JJ, PF, AK and KD conduct the experiment, MC, XG, JS, AP and UB were the supervisors. All contributed to the corrections of the paper.

**Competing interests.**

The authors declare that they have no conflict of interest.

**Acknowledgements.**

This study was supported by the Swiss National Science Foundation (BSSGI0_155846, 200021_169787), the International ST cooperation program of China (2014DFA90780) and National natural science foundation of China (21976093). Carlo Bozzetti is acknowledged for SoFi training.



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





**Figures**

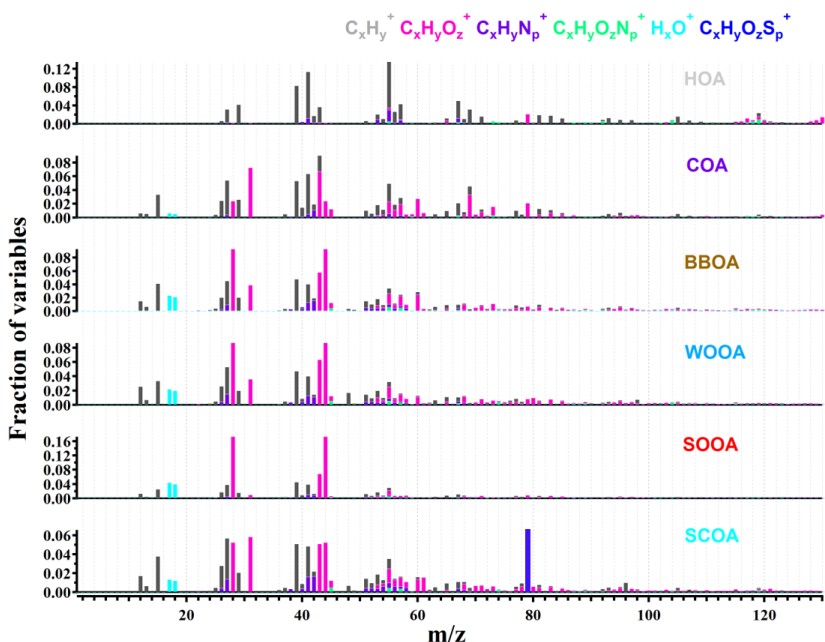

Fig.1 Factor profiles for the six-factor solution for AMS results with $HOA_{AMS}$ and $COA_{AMS}$ constrained by *a*=0.1. The total signal
5   of each factor is normalized to unity, and the y-axis presents the fractional contributions of the variables to the total signal of the
factor. ($HOA_{AMS}$: Hydrocarbon OA, $COA_{AMS}$: Cooking-related OA, $BBOA_{AMS}$: Biomass burning OA, $WOOA_{AMS}$: winter
oxygenated OA, $SOOA_{AMS}$: summer oxygenated OA, $SCOA_{AMS}$, sulfur-containing oxygenated OA).





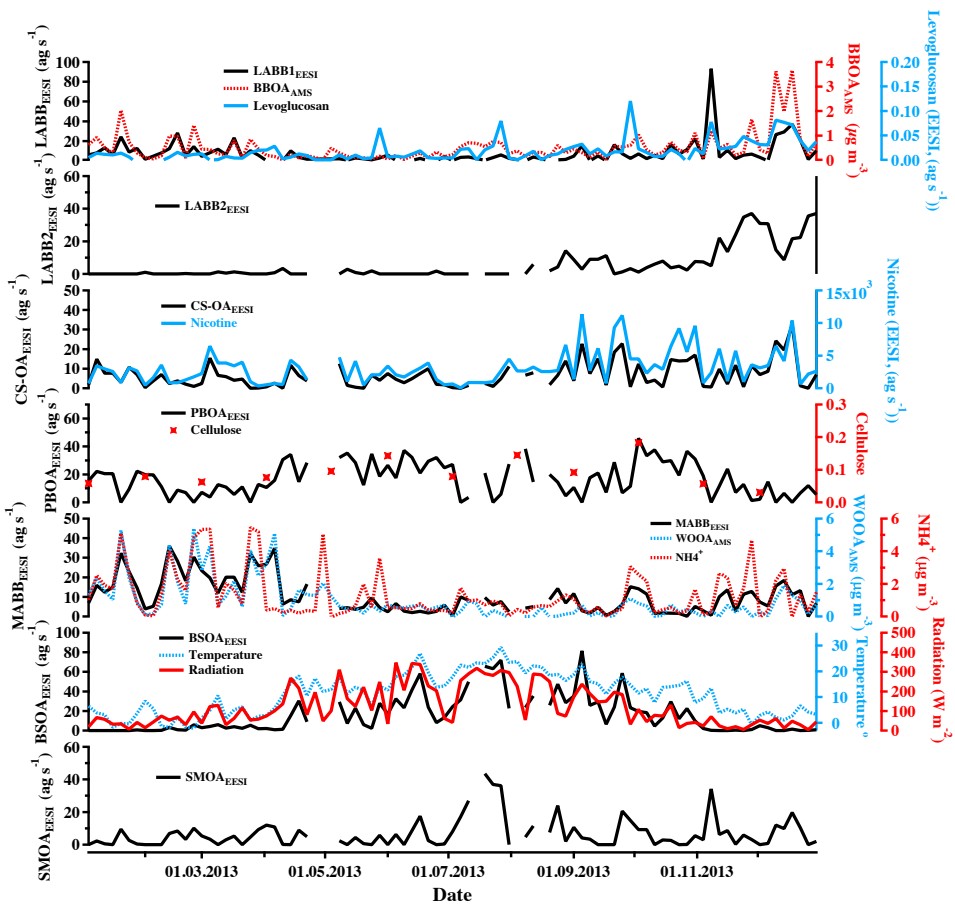

a)



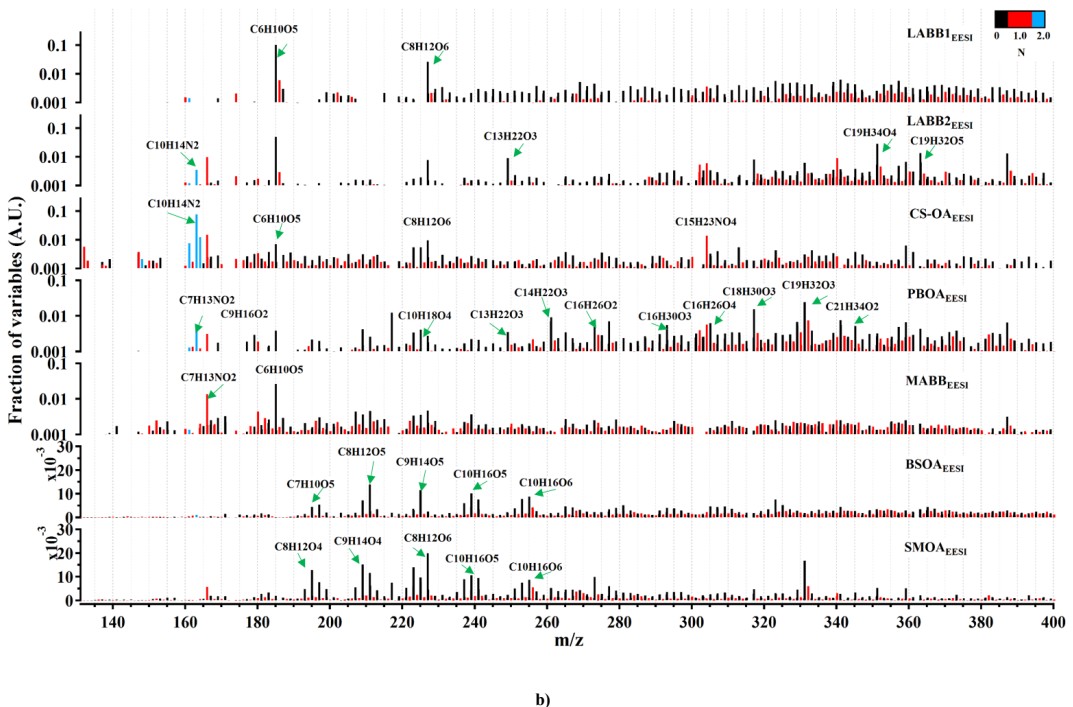

b)

**Fig. 2. (a) Time series of the EESI-TOF PMF analysis for the 7-factor solution, along with ancillary data. (b), Corresponding factor profiles. For all y-axes, EESI-TOF data are shown as mass flux (ag s$^{-1}$), AMS data are shown in µg m$^{-3}$, and other units are given.**



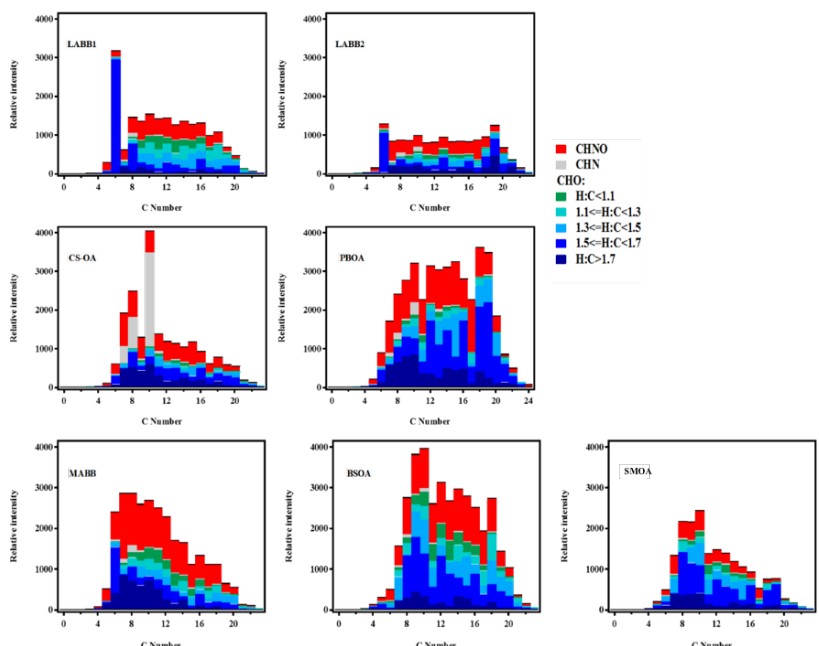

**Fig. 3 Stacked histogram binned by carbon number of ions, showing the apportioned intensity of each bin to each factor. Colors correspond to 7 families.**

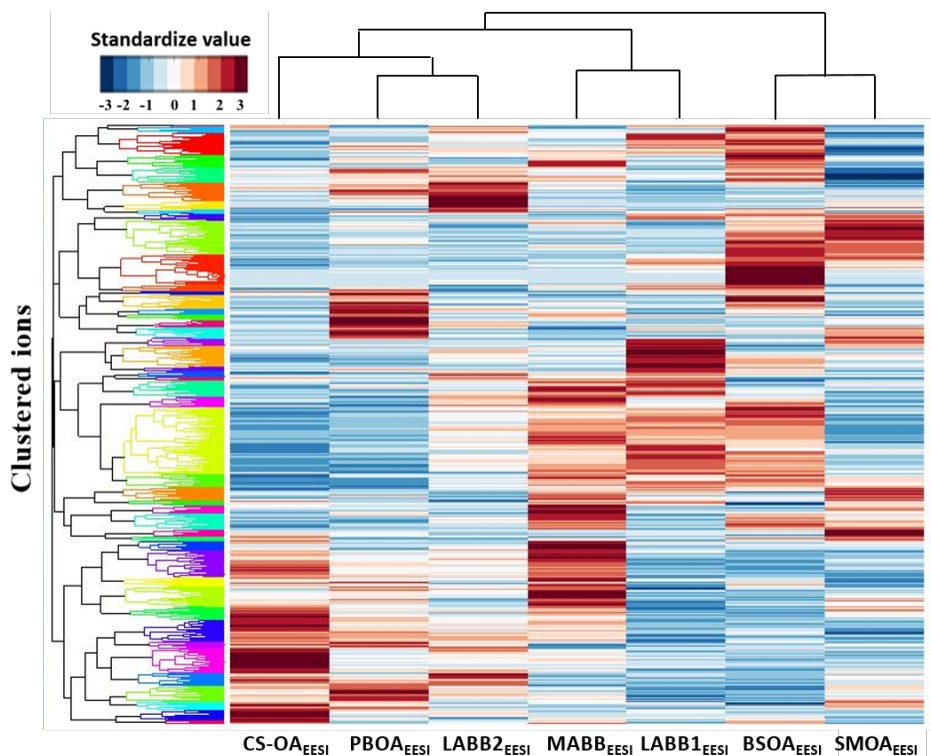

**Fig. 4 Standardized matrix of individual EESI-TOF ions vs. EESI-TOF PMF factors, colored by *z*-score. Ions and factors are sorted according to the results of their respective hierarchical clustering analysis; the resulting dendrograms are shown on the respective axes. The color of the compounds' groups in the dendrogram are chosen to make groupings convenient to read (dendrogram colors are chosen arbitrarily to aid the eye).**

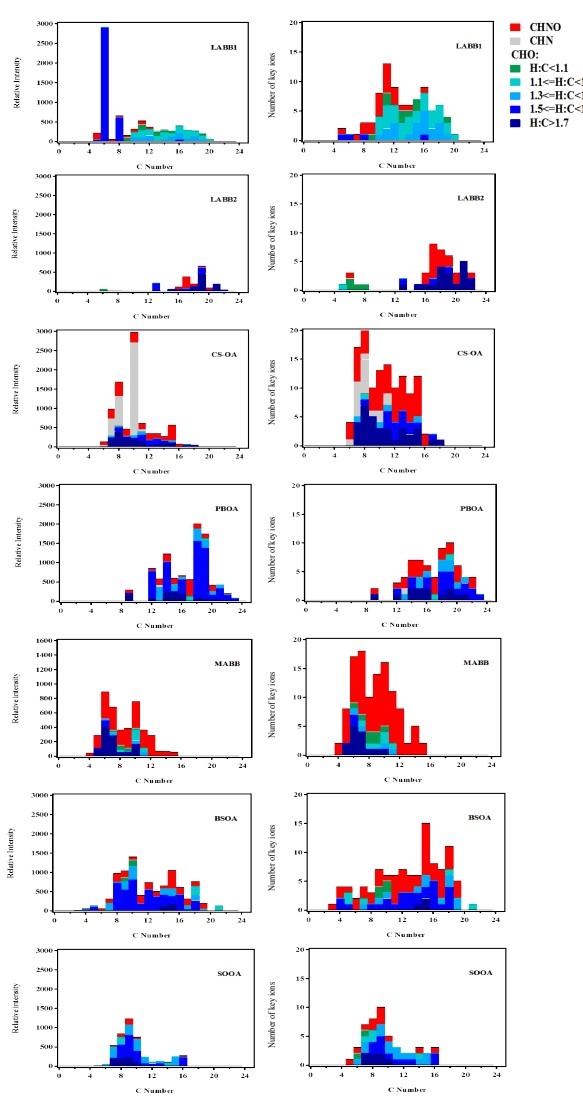

**Fig. 5 Stacked histogram binned by carbon number and colored by chemical family of key ions derived from clustergram analysis of factor mass spectra. Two representations are shown, with the stacked height denoting ion intensity (left column) or number of identified ions (right column).**



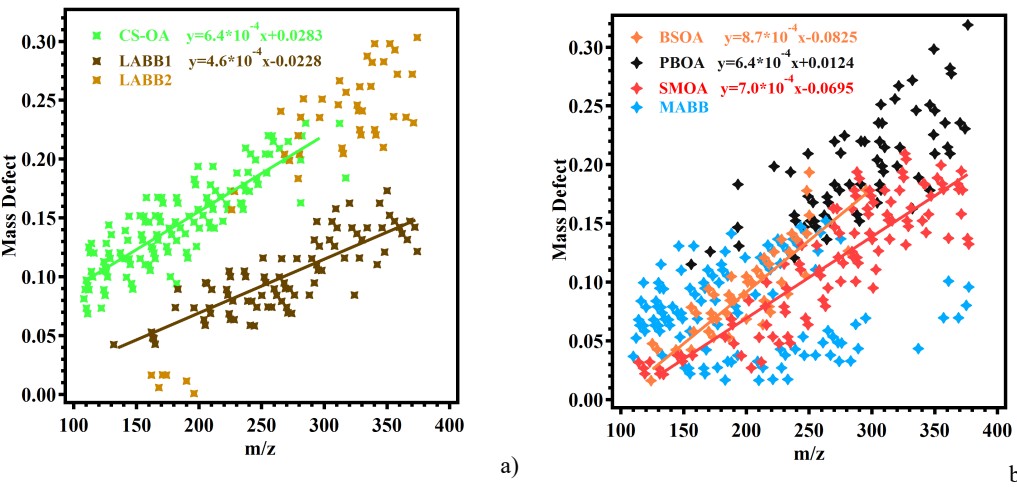

a)

b)

**Fig. 6 Mass defect plots of factor-specific ions (identified from the cluster analysis) for selected EESI-TOF POA (a) and SOA (b) factors.**





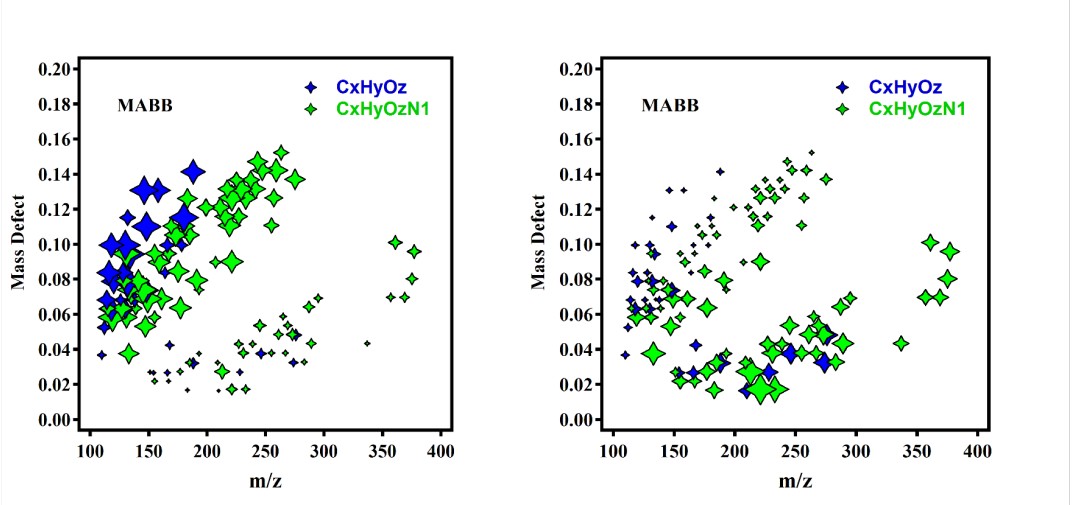

Fig. 7 Mass defect plot of factor-specific ions for the MABB$_{EESI}$ factor colored by nitrogen number, sized by H:C ratio (left) and O:C ratio (right).



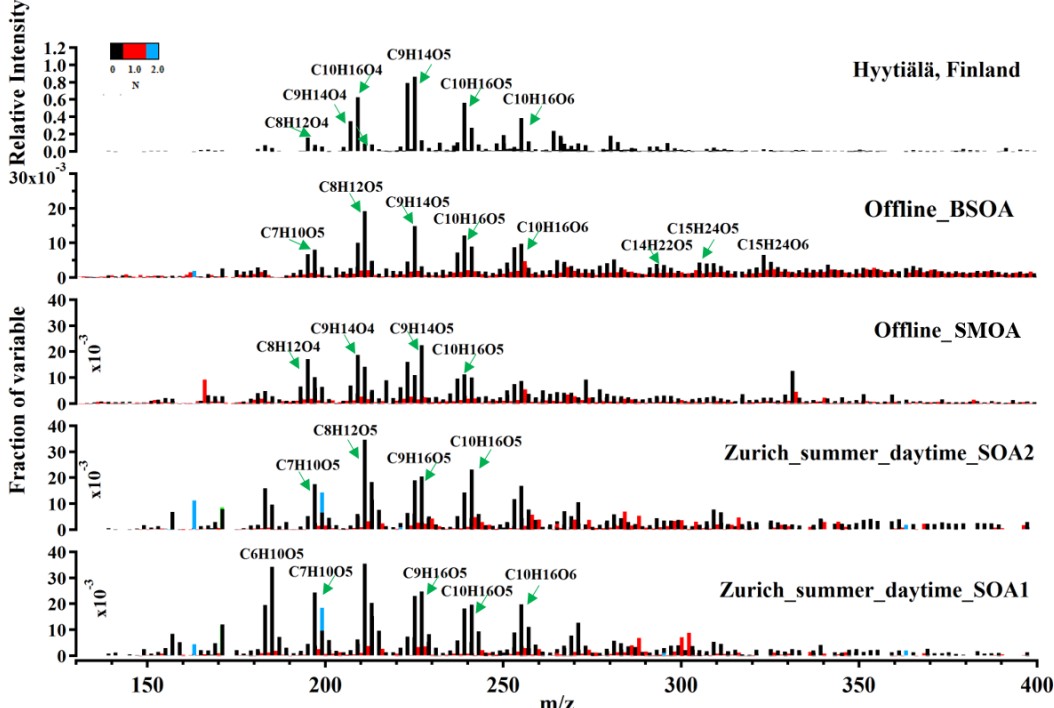

**Fig. 8 Comparison of the offline summer factor profiles with mass spectra from the Hyytiala field campaign and the online summer factors. The total signal of each factor is normalized to unity.**





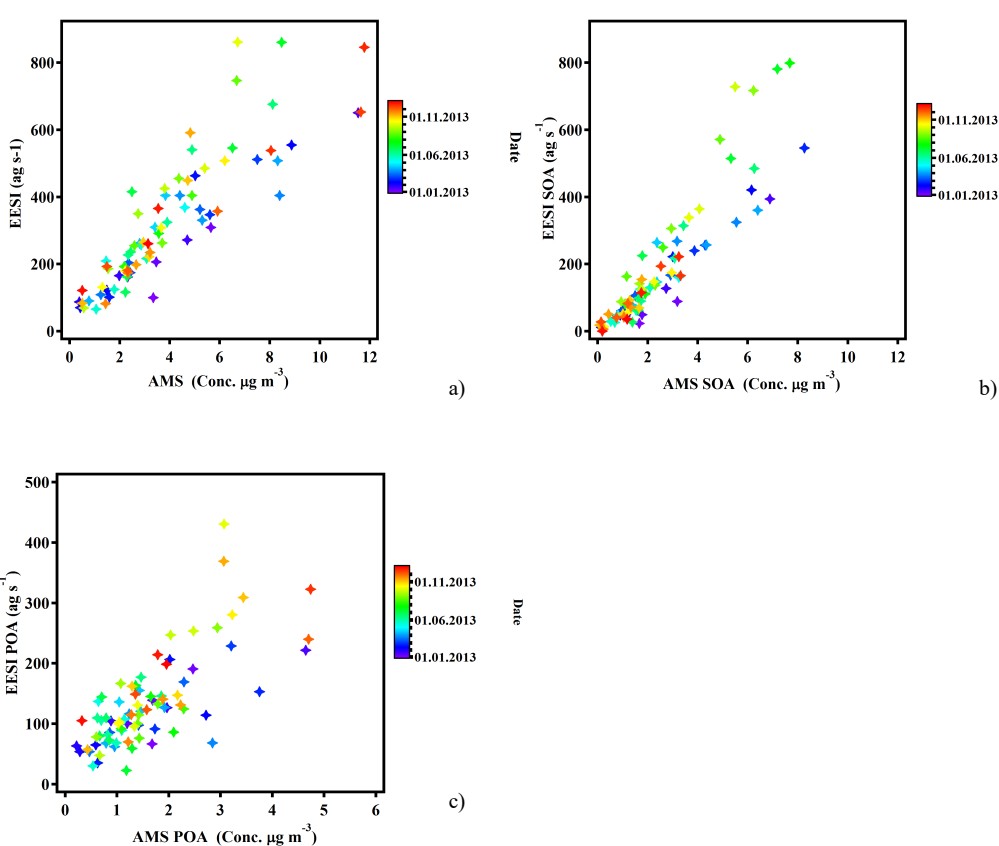





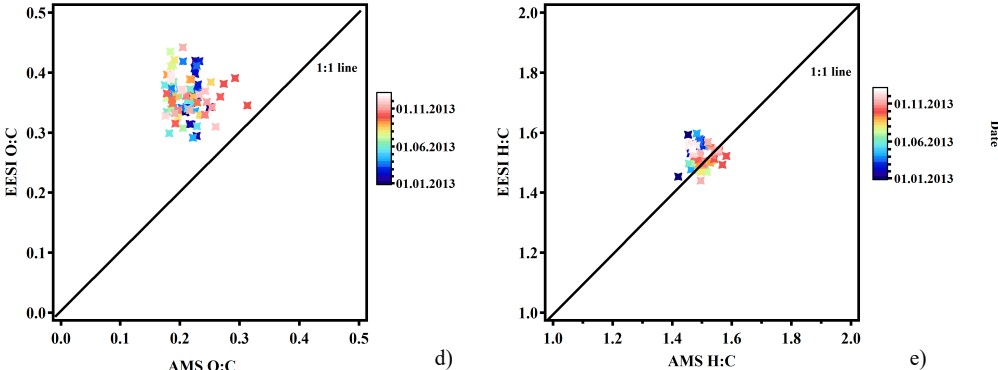

**Fig. 9 Comparison of EESI-TOF and AMS. Total EESI-TOF mass flux (ag s$^{-1}$) as a function of AMS OA, points are colored by date (a); total EESI-TOF SOA mass flux (ag s$^{-1}$) as a function of AMS SOA, points are colored by date (b); total EESI-TOF POA mass flux (ag s$^{-1}$) as a function of AMS POA, points are colored by date (c); The EESI-TOF and AMS comparison in terms of O:C (d) and H:C (e), points are colored by date.**





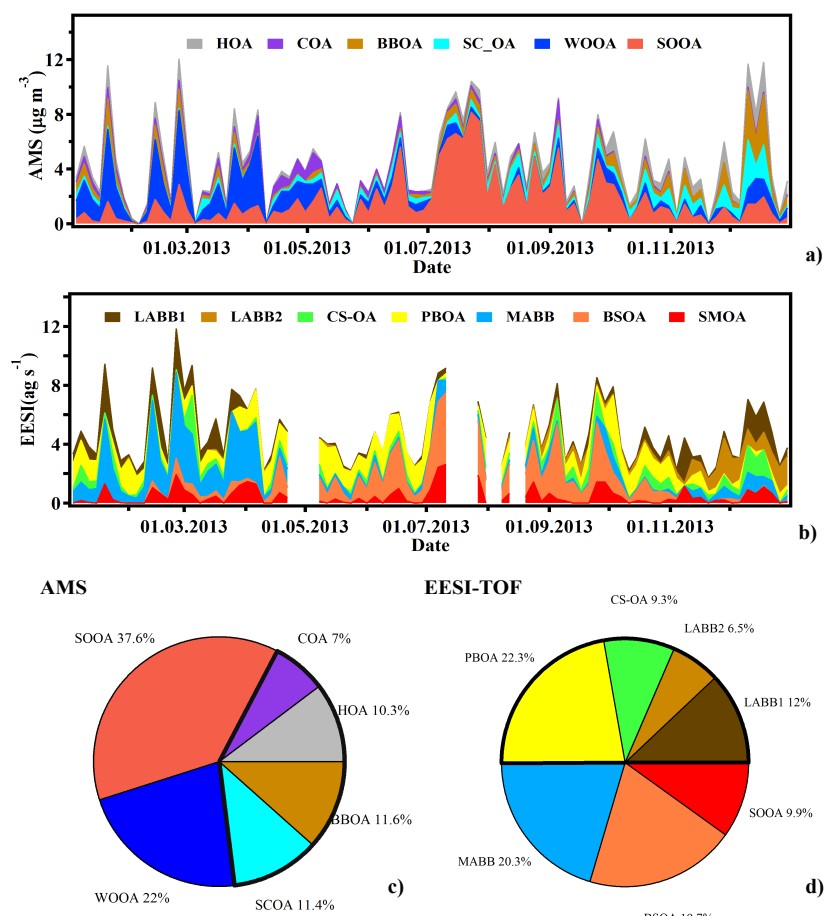

Fig. 10 Comparison between AMS factors and EESI factors: time series of the concentrations AMS PMF factor (a) and mass flux of EESI PMF factor (b). Pie charts of source apportionment results from the EESI (c) and AMS (d).

