# Peer review of "One-year characterization of organic aerosol composition and sources using an extractive electrospray ionization time-of-flight mass spectrometer (EESI-TOF)"

_Atmospheric Chemistry and Physics, 2019_

## Referee Comment (RC1) · Anonymous Referee #1 · 18 Feb 2020

The presented work is quite innovative, could be helpful for identification of the key sources over a long time period based on offline, highly chemically-resolved data provided by an EESI-TOF.

In the offline AMS section, line 20, Authors mentioned about WSOM fraction. What is the fraction of water non soluble organics? If the fraction is large, then it is important to account water non soluble fraction otherwise they will lead to error in the source contributions.

[Figure]

In the description of primary biological organic aerosol (PBOAEESI), author suggested that PBOA is not related to the cooking emissions. It was also mentioned that on comparison with previously obtained EESI-TOF COA factor, the dominant PBOAEESI ions are different from the major components of cooking-related EESI-TOF factors obtained from source apportionment of online summer and winter mass data. Also mentioned, the time series of the PBOAEESI and COAAMS factors are not well correlated. It would be nice to add the relevant profile of these factors in the supplementary.

Fig5, font size of the legends should be increased, not readable.

―――――――――――――――――――――――

---

## Referee Comment (RC2) · Anonymous Referee #2 · 24 Feb 2020

In this work, the authors used a novel offline analytical technique (EESI-TOF) to measure chemical components in organic aerosol. They adapted the online version of this technique to analyze filter extracts, allowing them to analyze filter samples without having to deploy an expensive instrument in the field. The time series of the chemical components are used to perform source apportionment by positive matrix factorization (PMF), a now standard procedure for statistical analyses of large data sets. They report a number of factors, and the interpretation of these factors is carefully discussed. This work is novel and potentially impactful, as this method will allow for widespread use of

a newly developed technique to measure a much larger number of organic compounds than previous techniques. I have one major concern that the authors need to address and a number of minor comments. If this concern can be satisfactorily addressed, I would recommend publication in ACP.

Major comments:

My only major question surrounds the ability to draw meaningful conclusions from the PMF results. In a previous paper, the authors found that the sensitivities of EESI-TOF for individual compounds vary by orders of magnitude and we do not yet have a good understanding of what determines the sensitivity. Here the authors are performing PMF and apportioning the signal of EESI-TOF to various factors, rather than the mass of OA. This is fundamentally different from PMF using AMS because AMS is much more quantitative. One could argue that these results could be biased by compounds to which EESI-TOF is very sensitive. For example, the fact that CSOA is so prominent is because EESI-TOF is very sensitive to tobacco smoke compounds like nicotine, but in reality CSOA is only a minor fraction of actual OA. I would like to see some discussion around this limitation, and how the authors can use these results to convince the readers that the source apportionment is representative. I would also note that the authors present convincing data in previous work that the signals are linear with mass, but that is not necessarily sufficient to avoid the bias described here.

Minor/technical comments:

I understand that using water as a solvent is useful in defining the observed mixture as WSOM. But would there be any utility to using a slightly more organic solvent for extraction such that a larger fraction of the OA can be recovered? The EESI is already using a water/acetonitrile mixture. Seems to me that using the same solvent mix could provide more insights.

A general but minor comment: what is the fraction of OM in PM10 observed here? Would be a good piece of information.

Abstract: In general, the abstract might be too specific, with too much reference into details. I would recommend cutting down on acronyms/abbreviations and focus on the more generalizable points. For example, in line 11 NABEL is not defined, nor is used in the rest of the abstract. This information is not necessary for a casual reader to grasp the main points of the manuscript

Abstract line 5: destroy is not an appropriate word. I recommend replace with "limits" Introduction Line 26: it is customary to include references for climate and health impacts of aerosols.

Introduction Line 31: POA sources and SOA mass do not seem to be parallel to each other. PMF apportions the mass to different sources

Page 3 lines 8-12: these arguments are valid, but filter sampling is also not typically operated in a dense network, because they require intense labor for both sample collection and extraction. Maybe it would be better to argue that since the filter sampling infrastructure is already in place, the marginal cost of additional sample analysis by AMS and/or EESI-TOF is low. Another advantage may be doing retrospective analysis from archived filters.

Page 3 line 23: same comment as above

Page 4 line 14: 14.7 cm diameter is not a standard size; is this correct? Usually 47mm diameter is used

Page 4 line 15: might be useful to add details of the PM10 sampler

Page 4 line 30: why is a temperature of 60C used? Seems rather high. Is it ensure evaporation of water? Is this the temperature required for EESI?

Page 5 line 2: are the mass spectral data averaged over the 5 minutes? Or some time point within those 5 minutes chosen for best representation of the extract?

Page 6 line 5: why is it necessary to use the mass? Isn't it sufficient to use ions (which

represents number of moles)?

Page 6 line 7: typo in "united", maybe "in units of"; same line: x represents the ion, not the composition

Page 6 line 14: unmatched parenthesis

Page 6 line 22: How is WSOC analyzed? Is it on the same Sunset OC-EC analyzer? Or is a different TOC analyzer? The way the first few sentences are written is vague and confusing.

Page 7 line 25: Which anchor profiles are used in this work? I did not really see the details clearly from the rest of the manuscript.

Page 8 line 28: there seems to be a typo here for the factor. The same problem is happening in other parts of the manuscript

Section 3.2.1: this is a very well written section with thoughtful support of interpretation.

Section 3.2.2: this section might be too detailed and only the last paragraph is relevant and interesting. I would recommend shortening the first few paragraphs significantly or move the details to figure captions/supplementary information. I was very distracted by much of the detailed discussion.

Page 11 lines 15-20: might be useful to add a short explanation (in SI) of how mass defect works and the trends in mass defect support structure identification. (e.g. why is the theoretical slope 6e-4?

Page 11 lines 24-29: it is somewhat unusual to see a cigarette smoke factor being so dominant in PMF. Why is this different from previous AMS or molecular marker source apportionment analysis in urban areas? Is it because the EESI-TOF is more selective towards nitrogen-containing compounds? Or is it the size dependence? (PM10 vs PM2.5 or even PM1)

Page 12 Line 19 and Fig 2a: judging from the figure, they do not look correlated (the

"dynamic range" is quite narrow for the cellulose measurement.) This line of evidence is relatively weak.

Page 12 Line 23: EESI only suggest formulas consistent with fatty acids, and are not definitively identifying them as fatty acids.

Page 12 Line 26-27: if low solubility of fatty acids for EESI-TOF is invalidating the COA identification, wouldn't that also refute the identification of plant debris as well? The formulas suggested in the beginning of this discussion are similar to those in COA, and I do not expect their (water) solubility to be significantly higher.

Page 13 lines 24-26: are any of the phenolic products from lignin pyrolysis observed?

Page 13 lines 27 – page 14 line 4: the paragraph focuses on arguing these two factors are observed separately (i.e. resolved by EESI-TOF and PMF), but a discussion of what are the actual difference of the sources would be useful and is currently lacking. What might be responsible for the difference (e.g. fuel type, residential vs wildfire, burning conditions etc.)?

Page 14 line 31: not sure what purpose the equation serves. It does not necessarily show that the dependence is exponential. Might be better to show a log-linear plot (in SI).

Page 15 line 10: I do not see convincing evidence that terpene is necessarily the precursor for SMOA. I am also confused by what do the authors mean by "fragmentation products from terpene oxidation ... followed by condensation after oxidation of light aromatics via ring-opening". This needs to be clarified.

Page 16 line 23: the number of filters is 86 here, but 91 earlier. Check for consistency.

---

## Author Comment (AC1) · 17 May 2020

**Response to the comments of anonymous referee #1**

We thank the referee for the valuable comments which have greatly helped us improve the manuscript. Please find below our responses (in black) after the referee comments (in blue). The changes in the revised manuscript are written in *italic*.

1.In the offline AMS section, line 20, Authors mentioned about WSOM fraction. What is the fraction of water non- soluble organics? If the fraction is large, then it is important to account water non-soluble fraction otherwise they will lead to error in the source contributions.

Response: On average, WSOM comprised 58% of the total OM. The relevance of the WSOM source apportionment to the total OM was addressed in detail in previous manuscripts discussing the offline AMS source apportionment for this dataset (Daellenbach et al., 2016; 2017) and is not repeated here; we focus instead on the comparison of AMS/EESI-TOF WSOM results. Individual AMS factors were found to have factor recoveries ranging from 0.54 to 0.89, with the exception of traffic-related hydrocarbon-like organic aerosol (HOA), which is not efficiently detected by the ionization technique used by the EESI-TOF in the present study. However, because the EESI-TOF and AMS here measure the same extracts, the AMS-determined recoveries can be assumed to apply to both instruments.

The WSOM fraction is now reported in the manuscript (page/line) as follows: *"Here we summarize the results of the AMS-PMF analysis on the WSOM fraction comprised 58 % of the total OM, which as noted in Section 2.2 are very similar to those of Daellenbach et al. (2017), conducted on different extracts from the same ambient filter samples."*

2.In the description of primary biological organic aerosol (PBOA$_{EESI}$), author suggested that PBOA is not related to the cooking emissions. It was also mentioned that on comparison with previously obtained EESI-TOF COA factor, the dominant PBOA$_{EESI}$ ions are different from the major components of cooking-related EESI-TOF factors obtained from source apportionment of online summer and winter mass data. Also mentioned, the time series of the PBOA$_{EESI}$ and COA$_{AMS}$ factors are not well correlated. It would be nice to add the relevant profile of these factors in the supplementary.

Response: As suggested, the comparison between online EESI-TOF COA (summer and winter) and offline PBOA mass spectra is added in the supplement. It's quite clear that these two factors have different features.;

[Figure]

3.Fig5, font size of the legends should be increased, not readable.

Response: Revised. The legends could be read now.

---

## Author Comment (AC2) · 17 May 2020

**Response to the comments of anonymous referee #2**

We thank the referee for the valuable comments which have greatly helped us improve the manuscript. Please find below our responses (in black) after the referee comments (in blue). The changes in the revised manuscript are written in *italic*.

Major comments:

1.My only major question surrounds the ability to draw meaningful conclusions from the PMF results. In a previous paper, the authors found that the sensitivities of EESI-TOF for individual compounds vary by orders of magnitude and we do not yet have a good understanding of what determines the sensitivity. Here the authors are performing PMF and apportioning the signal of EESI-TOF to various factors, rather than the mass of OA. This is fundamentally different from PMF using AMS because AMS is much more quantitative. One could argue that these results could be biased by compounds to which EESI-TOF is very sensitive. For example, the fact that CSOA is so prominent is because EESI-TOF is very sensitive to tobacco smoke compounds like nicotine, but in reality CSOA is only a minor fraction of actual OA. I would like to see some discussion around this limitation, and how the authors can use these results to convince the readers that the source apportionment is representative. I would also note that the authors present convincing data in previous work that the signals are linear with mass, but that is not necessarily sufficient to avoid the bias described here.

Response: The reviewer is correct that the uncertainties in the relative sensitivity of the EESI-TOF pose a challenge for interpretation of source apportionment results. This was also discussed previously in online EESI-TOF source apportionment analyses (Stefenelli et al., 2019; Qi et al, 2019). Empirically, the EESI-TOF source apportionment results obtained in these earlier studies are in good agreement with the AMS, with the exception of factors containing a high fraction of levoglucosan, for which the EESI-TOF sensitivity is known to be high (Lopez-Hilfiker et al., 2019). A similar analysis is the subject of section 3.4, and is again important for constraining the effects of compound-dependent sensitivities on EESI-TOF source apportionment. This point is clarified in the manuscript as follows:

*"The relatively good agreement between related factors across instruments suggests that compound-dependent sensitivities are not resulting in a major distortion of the EESI-TOF source apportionment results."*

Minor/technical comments:

2.I understand that using water as a solvent is useful in defining the observed mixture as WSOM. But would there be any utility to using a slightly more organic solvent for extraction such that a larger fraction of the OA can be recovered? The EESI is already using a water/acetonitrile mixture. Seems to me that using the same solvent mix could provide more insights.

Response: We agree that the use of organic or mixed water/organic solvents for extraction is potentially of high interest. In the present study, we utilized water primarily because it has been used in several previous offline AMS analyses (Daellenbach et al., 2016, 2017; Bozzetti et al., 2017; Vlachou et al., 2018;), and thus represents a wellcharacterized technique against which the new EESI-TOF measurements can be compared. This initial validation can then be a stepping-stone to future investigations of less investigated extraction techniques.

A general but minor comment:

3.what is the fraction of OM in PM10 observed here? Would be a good piece of information.
The WSOM fraction is now reported in the manuscript (page/line) as follows: *"Here we summarize the results of the AMS-PMF analysis on the WSOM fraction comprised 58 % of the total OM, which as noted in Section 2.2 are very similar to those of Daellenbach et al. (2017), conducted on different extracts from the same ambient filter samples."*

4.Abstract: In general, the abstract might be too specific, with too much reference into details. I would recommend cutting down on acronyms/abbreviations and focus on the more generalizable points. For example, in line 11 NABEL is not defined, nor is used in the rest of the abstract. This information is not necessary for a casual reader to grasp the main points of the manuscript

Response: We agree with the comment. Some detailed / abbreviations are not needed in abstract. Now we revised to *"Here, we apply the same strategy for EESI-TOF measurements of 1 year of 24-hour filter samples collected approximately every 4th day throughout 2013 at an urban site. The nebulized water extracts were measured simultaneously with an AMS. The application of positive matrix factorization (PMF) to EESI-TOF spectra resolved seven factors, which describe water-soluble OA: less and more aged biomass burning aerosol (LABB$_{EESI}$ and MABB$_{EESI}$, respectively), cigarette smoke-related organic aerosol, primary biological organic aerosol, biogenic secondary organic aerosol, and a summer mixed oxygenated organic aerosol factor. Seasonal trends and relative contributions of the EESI-TOF OA sources were compared with AMS source apportionment factors, measured water-soluble ions, cellulose, and meteorological data. Cluster analysis was utilized to identify key factor-specific ions based on PMF. Both LABB and MABB contribute strongly during winter. LABB is distinguished by very high signals from $C_6H_{10}O_5$ (levoglucosan and isomers) and $C_8H_{12}O_6$, whereas MABB is characterized by a large number of CxHyOz and CxHyOzN species two distinct populations: one with low H:C and high O:C, and the other with high H:C and low O:C. Two oxygenated summertime SOA sources were attributed to terpene-derived biogenic SOA, a major summertime aerosol source in Central Europe. Furthermore, a primary biological organic aerosol factor was identified, which was dominated by plant-derived fatty acids and correlated with free cellulose. The cigarette smoke-related factor contained a high contribution of nicotine and high abundance of organic nitrate ions with low m/z."*

5.Abstract line 5: destroy is not an appropriate word. I recommend replace with "limits"

Response: Revised.

6.Introduction Line 26: it is customary to include references for climate and health impacts of aerosols.

Response: Two references are added: *"Heal et al., 2012; Kelly et al., 2012.*

*Heal, M. R., Kumar, P., and Harrison, R. M.: Particles, air quality, policy and health, Chem. Soc. Rev., 41, 6606-6630, 10.1039/c2c35076a, 2012.*

*Kelly, F. J., Fussell, J. C.: Size, source and chemical composition as determinants of toxicity attributable to ambient particulate matter, Atmos. Env., 60, 504-526, 10.1016/j. atmosenv. 2012. 06. 039, 2012."*

7.Introduction Line 31: POA sources and SOA mass do not seem to be parallel to each other. PMF apportions the mass to different sources.

Response: With rare exceptions (e.g. IEPOX-derived SOA), AMS source apportionment analyses represent SOA as a linear combination of less- and more-oxygenated oxygenated organic aerosol factors (LO-OOA and MO-OOA, respectively). While this provides quantification of the total SOA fraction and a means to describe bulk SOA compositional variabilty (e.g. atomic O:C ratio), it does not provide direct links to SOA sources. We have clarified this point in the text as follows:

*"The Aerodyne aerosol mass spectrometer (AMS) provides online measurements of OA composition and in combination with statistical methods such as positive matrix factorization (PMF) has greatly advanced the quantification of primary organic aerosol (POA) sources and total secondary organic aerosol (SOA) mass, although individual SOA sources are not typically separable."*

8.Page 3 lines 8-12: these arguments are valid, but filter sampling is also not typically operated in a dense network, because they require intense labor for both sample collection and extraction. Maybe it would be better to argue that since the filter sampling infrastructure is already in place, the marginal cost of additional sample analysis by AMS and/or EESI-TOF is low. Another advantage may be doing retrospective analysis from archived filters.

Response: Revised: *"(5) The expense of additional sample analysis by new developed instruments is low once the filter sampling infrastructure is installed."*

9.Page 3 line 23: same comment as above

Response: This issue was addressed in response to comment (7). We feel the text modification in response to the earlier comment clarifies the point and leave the text in this section unchanged.

Page 4 line 14: 14.7 cm diameter is not a standard size; is this correct? Usually 47mm diameter is used

Response: We use the same filters as in Daellenbach et al. (2016; 2017). The filters are 15 cm but the exposed area has a diameter of 14.7 cm.

Page 4 line 15: might be useful to add details of the PM10 sampler

Response: The manufacturer and model are added in manuscript: "*Digitel DHA80*".

Page 4 line 30: why is a temperature of 60C used? Seems rather high. Is it ensure evaporation of water? Is this the temperature required for EESI?

Response: This is the temperature for the drying unit of the Apex nebulizer and is also used in other offline AMS studies. It is not specific to the EESI-TOF

Page 5 line 2: are the mass spectral data averaged over the 5 minutes? Or some time point within those 5 minutes chosen for best representation of the extract?

Response: Sorry for the confusion. Five minutes / ten minutes switching is for aerosol generation from the filter extract / milli-Q water. EESI-TOF mass spectra are recorded in positive mode at 5 s time resolution in these 5 minutes (in the section for the instrument, Page 6, line 2), then time points (except the first minute points) for each filter are chosen and averaged for the PMF.

Page 6 line 5: why is it necessary to use the mass? Isn't it sufficient to use ions (which represents number of moles)?

Response: This conversion is performed to optimize the comparison with the AMS, which is a mass-based measurement. Although the comparison remains imperfect, as it neglects uncertainties in EESI-TOF ion transmission and ion-specific sensitivity, it is a step closer than mass vs. moles.

Page 6 line 7: typo in "united", maybe "in units of"; same line: x represents the ion, not the composition

Response: Revised. "*Here Mx is the mass flux of ions in unit of in ag s-1, where x represents the measured molecular ion.*"

Page 6 line 14: unmatched parenthesis

Response: Revised. "*Note that because online EESI-TOF operation already requires extraction into the spray droplets (1:1 water:acetonitrile mixture), that major differences between the measured OA fraction between online and offline analyses are unlikely.*"

Page 6 line 22: How is WSOC analyzed? Is it on the same Sunset OC-EC analyzer? Or is a different TOC analyzer? The way the first few sentences are written is vague and confusing.

Response: WSOC is analyzed using a different TOC analyzer. Here we revised the sentences, "*Filters were analyzed for organic and elemental carbon (OC, EC) using a thermo-optical transmission method with a Sunset OC-EC analyzer, following the EUSAAR-2 thermal-optical transmission protocol (Cavalli et al., 2010). Water-*

*soluble organic carbon was measured using a total organic carbon analyzer with water extraction followed by catalytic oxidation and nondispersive infrared detection of CO2."*

Page 7 line 25: Which anchor profiles are used in this work? I did not really see the details clearly from the rest of the manuscript.

Response: The details is described in section 3.1, Page 8, line 20, *"HOA$_{AMS}$ and COA$_{AMS}$ mass profiles were constrained using anchor profiles obtained from winter in Paris (Crippa et al., 2013b) with a-values of 0.1 and 0.2, respectively."*

Page 8 line 28: there seems to be a typo here for the factor. The same problem is happening in other parts of the manuscript

Response: Revised. *"The oxygenated OA factors are resolved based on the differences in their seasonal behavior: SOOA$_{AMS}$ (elevated in summer) and WOOA$_{AMS}$ (elevated in winter)."*

Section 3.2.1: this is a very well written section with thoughtful support of interpretation.

Section 3.2.2: this section might be too detailed and only the last paragraph is relevant and interesting. I would recommend shortening the first few paragraphs significantly or move the details to figure captions/supplementary information. I was very distracted by much of the detailed discussion.

Response: We have removed discussion of the supplementary figures and slightly condensed the text. However, we do consider the remaining information as crucial to understanding the detailed factor discussion that follows, as this text introduces the key figures and concepts crucial to that discussion.

Page 11 lines 15-20: might be useful to add a short explanation (in SI) of how mass defect works and the trends in mass defect support structure identification. (e.g. why is the theoretical slope 6e-4?)

Response: We have clarified the source of the theoretical slopes for CH and CHO as follows:

*"The slope for the LABB1$_{EESI}$ factor of 4.6*10$^{-4}$ is consistent with addition of CH groups, which yield a slope of 6*10$^{-4}$ (mass of CH is 13.00783, so slope corresponds to mass defect of 0.00783 added over 13 m/z)."*

*"Both the slopes of BSOA$_{EESI}$ (8.7*10$^{-4}$) and SMOA$_{EESI}$ (7.0*10$^{-4}$) are consistent with the addition of CHO functional groups, which would yield a slope of 1*10$^{-3}$ (mass defect of 0.00274 over 29 m/z)."*

Page 11 lines 24-29: it is somewhat unusual to see a cigarette smoke factor being so dominant in PMF. Why is this different from previous AMS or molecular marker source apportionment analysis in urban areas? Is it because

the EESI-TOF is more selective towards nitrogen-containing compounds? Or is it the size dependence? (PM10 vs PM2.5 or even PM1)

Response: EESI-TOF has the advantage to identify molecular ions, which means that signal from nicotine is exclusively contained in this unique ion. In contrast, while the AMS does contain a proposed tracer for nicotine in the $C_5H_{10}N^+$ (Struckmeier et al., 2016; Qi et al., 2019), this ion represents only a minor fraction of the nicotine mass spectrum and in ambient aerosol may be difficult to resolve from other nearby species. While the sensitivity of the EESI-TOF to nicotine has not been well-characterized, the good agreement between AMS and EESI-TOF source apportionment of CS-OA for online measurements at the same site in summer (Stefenelli et al., 2019) suggests this is not the dominant cause. The prevalence of CS-OA in the present study may be influenced by the specific sampling location, which is surrounded by restaurants, and thus likely subject to high local smoking activity.

Page 12 Line 19 and Fig 2a: judging from the figure, they do not look correlated (the "dynamic range" is quite narrow for the cellulose measurement.) This line of evidence is relatively weak.

Response: Cellulose and PBOA$_{EESI}$ are relatively well correlated, with $R$ = 0.83. This is much higher than the correlation of cellulose with other EESI-TOF factors, for which the $R$ are all less than 0.3, which suggests the correlation is meaningful. . The contrast between the cellulose/PBOA$_{EESI}$ correlation and the other factors has been added to the text as follows:
*"As shown in Fig. 2a, cellulose correlates with the PBOA$_{EESI}$ time series (R=0.83) much more strongly than with any other EESI-TOF factor (R < 0.3)"*

Page 12 Line 23: EESI only suggest formulas consistent with fatty acids, and are not definitively identifying them as fatty acids.

Response: The original text did not clearly distinguish between expected source composition (where some molecules are identified) and the EESI-TOF profiles (where only ion formulas are known). The revised text reads as follows:
Revised: *"Viewed broadly, these two emissions sources are somewhat similar in that they are both expected to have strong contributions from fatty acids, consistent with the salient features of the PBOA$_{EESI}$ mass spectrum."*

Page 12 Line 26-27: if low solubility of fatty acids for EESI-TOF is invalidating the COA identification, wouldn't that also refute the identification of plant debris as well? The formulas suggested in the beginning of this discussion are similar to those in COA, and I do not expect their (water) solubility to be significantly higher.

Response: We agree that the fatty acids feature of PBOA factor mass spectrum is similar to those in COA and that their recovery is likely to be low. However, in the AMS PMF analysis, the COA factor profile was constrained using a reference spectrum from a previous study. This approach has been shown to improve resolution of lowintensity factors. This approach is not viable for EESI-TOF COA in the current study due to the low number of previous studies and study-to-study variation in COA factor profiles.

Page 13 lines 24-26: are any of the phenolic products from lignin pyrolysis observed?

Response: Ions consistent with syringic acid and vanillic acid are were measured in our study. These compounds are derived from the oxidation of lignin decomposition products, which in turn are a major component of biomass combustion emissions and apportioned primarily to aged biomass burning factors. Here, in LABB section, we don't include the analysis of these phenolic products.

Page 13 lines 27 – page 14 line 4: the paragraph focuses on arguing these two factors are observed separately (i.e. resolved by EESI-TOF and PMF), but a discussion of what are the actual difference of the sources would be useful and is currently lacking. What might be responsible for the difference (e.g. fuel type, residential vs wildfire, burning conditions etc.)?

Response: We agree that this would be a very interesting addition, but at present we are unfortunately not able to link these differences to specific specific sources and/or burning conditions. Given the sampling location, it is likely that residential combustion dominates the biomass burning emissions in winter and wildfires are unlikely to be significant, However, we cannot conclusively distinguish between factors such as fuel type, burning conditions, or stove age.

Page 14 line 31: not sure what purpose the equation serves. It does not necessarily show that the dependence is exponential. Might be better to show a log-linear plot (in SI).

Response: The log-linear plot is showed in Fig. 10, and the equation has been deleted.

Page 15 line 10: I do not see convincing evidence that terpene is necessarily the precursor for SMOA. I am also confused by what do the authors mean by "fragmentation products from terpene oxidation followed by condensation after oxidation of light aromatics via ring-opening". This needs to be clarified.

Response: SMOA likely includes oxidation products from both terpenes and light aromatics. This has been clarified in the text as follows:
*"The SMOA$_{EESI}$ factor mostly with less than 10 carbon atoms (C7, C8, C9 and C10) likely includes both monoterpene oxidation products (e.g., $C_{10}H_{16}O_4$, $C_{10}H_{16}O_5$) and ring-opening oxidation products of light aromatics."*

Page 16 line 23: the number of filters is 86 here, but 91 earlier. Check for consistency.

Response: Revised to 91 filters.